# Transcription elongation is finely tuned by dozens of regulatory factors

**Mary Couvillion[†], Kevin M Harlen[†], Kate C Lachance[†], Kristine L Trotta, Erin Smith, Christian Brion, Brendan M Smalec, L Stirling Churchman***

Blavatnik Institute, Department of Genetics, Harvard Medical School, Boston, United States

**Abstract** Understanding the complex network that regulates transcription elongation requires the quantitative analysis of RNA polymerase II (Pol II) activity in a wide variety of regulatory environments. We performed native elongating transcript sequencing (NET-seq) in 41 strains of *Saccharomyces cerevisiae* lacking known elongation regulators, including RNA processing factors, transcription elongation factors, chromatin modifiers, and remodelers. We found that the opposing effects of these factors balance transcription elongation and antisense transcription. Different sets of factors tightly regulate Pol II progression across gene bodies so that Pol II density peaks at key points of RNA processing. These regulators control where Pol II pauses with each obscuring large numbers of potential pause sites that are primarily determined by DNA sequence and shape. Antisense transcription varies highly across the regulatory landscapes analyzed, but antisense transcription in itself does not affect sense transcription at the same locus. Our findings collectively show that a diverse array of factors regulate transcription elongation by precisely balancing Pol II activity.

*For correspondence:
churchman@genetics.med.
harvard.edu

[†]These authors contributed
equally to this work

## Editor's evaluation

In this manuscript the authors have conducted native elongation transcript sequencing on yeast strains deleted for one of 41 different transcription, chromatin modifying and RNA processing factors. They find that a large fraction of these deletions affect transcription elongation and RNA Pol II pausing indicating that elongation is carefully regulated by many factors.

## Introduction

Transcription is a highly regulated and conserved process that consists of three phases: initiation, elongation, and termination (*Shandilya and Roberts, 2012*; *Svejstrup, 2004*). Post-initiation regulation is critical for co-transcriptional RNA processing, shaping the chromatin landscape, and preventing run-on transcription into downstream genes (*Herzel et al., 2017*; *Holmes et al., 2015*; *Proudfoot et al., 2002*; *Rando and Winston, 2012*). Transcription elongation is controlled across gene bodies by a wide variety of factors, including transcription factors, chromatin modifiers, chromatin assembly factors and chaperones, RNA processing factors, and histone variants. Understanding how these factors act separately and in concert to influence RNA polymerase II (Pol II) activity will shed light on how transcription elongation and co-transcriptional processes are coordinated.

Transcription is a discontinuous process: periods of productive elongation are frequently interrupted by pauses. Pol II pausing was first observed in vitro in *Escherichia coli* polymerase transcribing the *lac* operon and lambda DNA (*Dahlberg and Blattner, 1973*; *Gilbert et al., 1974*; *Kassavetis and Chamberlin, 1981*; *Kingston and Chamberlin, 1981*; *Lee et al., 1976*; *Maizels, 1973*). Observations of Pol II pausing in vivo provided the first evidence of promoter proximal pausing (*Gariglio et al., 1981*). These findings were extended by chromatin immunoprecipitation (ChIP) studies, which

identified paused polymerase near the 5' ends of certain *Drosophila* and mammalian genes (*Bentley and Groudine, 1986*; *Eick and Bornkamm, 1986*; *Gilmour and Lis, 1986*; *Krumm et al., 1992*; *Nepveu and Marcu, 1986*; *Rougvie and Lis, 1988*; *Spencer and Groudine, 1990*; *Strobl and Eick, 1992*).

The advent of high-throughput and high-resolution sequencing technologies has led to the development of sequencing methods such as NET-seq and precision run-on sequencing (PRO-seq) that measure Pol II density genome-wide at nucleotide resolution. Collectively, these techniques have highlighted the control of transcription elongation by regulatory factors. These approaches and other nascent RNA sequencing methods visualize the production of transcripts from RNA polymerases across the genome (*Churchman and Weissman, 2011*; *Core et al., 2008*; *Kwak et al., 2013*; *Mayer et al., 2015*; *Nojima et al., 2015*; *Schwalb et al., 2016*), and therefore are capable of revealing the immediate and direct effects of a perturbation on transcription. In addition, these assays capture unstable transcripts such as antisense RNAs, which can be critical to transcription regulation but are invisible by many techniques (*Camblong et al., 2007*; *Hongay et al., 2006*; *Martens et al., 2004*; *Uhler et al., 2007*). The strand-specificity and high resolution of these methods are transforming our understanding of transcription elongation and regulation.

NET-seq, PRO-seq, and other high-resolution methods have revealed both regions of high Pol II density, such as promoter proximal pausing, and specific sites of Pol II pausing across gene bodies (*Churchman and Weissman, 2011*; *Ferrari et al., 2013*; *Kindgren et al., 2020*; *Kwak et al., 2013*; *Larson et al., 2014*; *Mayer et al., 2015*; *Nojima et al., 2015*; *Vvedenskaya et al., 2014*; *Weber et al., 2014*). Regions or peaks of high Pol II density, such as promoter proximal pauses, are created in part by a high density of pause sites that together create barriers to elongation and provide an opportunity for regulation and coordination of co-transcriptional events (*Bentley, 2014*; *Mayer et al., 2017*; *Noe Gonzalez et al., 2021*; *Rougvie and Lis, 1988*). Myriad factors control Pol II peaks in vivo. For example, in yeast prominent peaks of Pol II density occur near polyadenylation [poly(A)] sites (*Harlen et al., 2016*). Loss of Rtt103, a termination factor, causes a dramatic peak in Pol II density directly downstream of poly(A) sites (*Harlen et al., 2016*). On the other hand, specific sites of Pol II pauses are reminiscent of pausing observed at single nucleotide positions in vitro (*Herbert et al., 2008*; *Kassavetis and Chamberlin, 1981*; *Kingston and Chamberlin, 1981*; *Mayer et al., 2017*). These in vitro pauses arise from intrinsic properties of the polymerase itself, interactions with the DNA template, and the presence of bound proteins (e.g. histones and transcription factors) (*Herbert et al., 2006*; *Hodges et al., 2009*; *Kassavetis and Chamberlin, 1981*; *Kireeva et al., 2005*; *Kireeva and Kashlev, 2009*; *Shaevitz et al., 2003*). NET-seq analysis of Pol II pause sites in yeast and mammalian cells has revealed a similar connection to DNA sequence and histones, but has not been explored across different regulatory landscapes (*Churchman and Weissman, 2011*; *Gajos et al., 2021*).

Pol II transcribes much of the genome in all eukaryotes, yet only a fraction of its transcripts mature into stable, protein-coding RNA products (*Bertone et al., 2004*; *Cheng et al., 2005*; *David et al., 2006*; *Hangauer et al., 2013*; *Kapranov et al., 2007*; *Mercer et al., 2011*; *Nagalakshmi et al., 2008*; *Smolle and Workman, 2013*; *Steinmetz et al., 2006*). A major contributor to unstable noncoding RNA products is antisense transcripts, i.e., RNAs transcribed from the strand opposite the sense strand of a protein-coding gene. Originally identified in bacteria (*Spiegelman et al., 1972*), antisense transcripts were soon discovered in eukaryotes as well (*Anderson et al., 1981*; *Bibb et al., 1981*). Since its discovery, antisense transcription has been detected opposite the vast majority of annotated genes in yeast (*Xu et al., 2011*), arising initially as a natural consequence of open chromatin regions (*Jin et al., 2017*). Antisense transcription regulates gene expression at a number of yeast genes (*Camblong et al., 2007*; *Hongay et al., 2006*; *Houseley et al., 2008*; *Lenstra et al., 2015*; *Martens et al., 2004*; *Uhler et al., 2007*); however, a general genome-wide function has not been identified (*Murray and Mellor, 2016*). To better understand pervasive antisense transcription and its role in regulation, it is important to determine whether it is tunable by regulatory factors, which would help distinguish whether the levels of antisense transcription are tightly set or whether antisense transcription is simply a nuisance that the cell works to minimize.

To gain insight into the regulation of the production of coding and non-coding transcripts by Pol II, we used NET-seq to analyze 41 *Saccharomyces cerevisiae* mutant strains lacking known elongation regulators. We investigated how each factor regulates nascent transcription, production of antisense transcripts, and pausing across gene bodies. Surprisingly, across these regulatory contexts, we find

that antisense transcription at a locus does not affect its sense transcription. Metrics describing each transcription phenotype span a broad dynamic range with wild-type activity lying near the center. The loss of each factor revealed distinct sets of pause sites that we used to create machine learning models of Pol II pausing, highlighting which genomic features can classify pause positions. Together, our results show that Pol II transcription elongation is determined by the contrasting impacts of many regulatory factors.

## Results

### Reverse genetic screen for transcription regulators

To obtain insight into the transcription regulatory network of *S. cerevisiae*, we individually deleted 41 non-essential transcription elongation regulators, including RNA processing factors, transcription elongation factors, histone variants, chromatin modifiers, and chromatin remodelers and chaperones, and assessed the transcriptional effects of each deletion using NET-seq (*Figure 1A*). The wild-type transcription baseline was established using four biological replicates of wild-type cells; the results from the replicates were highly correlated ($R^2 \geq 0.97$; *Figure 1—figure supplement 1A*). All mutant strains were analyzed in at least biological duplicate. Results from strain replicates were highly correlated ($R^2 \geq 0.75$; *Supplementary file 1*). Importantly, all replicates were performed at different times, by different researchers, and in different strain isolates, demonstrating the reproducibility of our results.

### Nascent gene expression is uniquely disrupted across deletion strains

Because all of the factors examined in our screen play roles in transcription regulation, we first sought to determine whether each factor regulates different sets of genes, or whether modifications of the transcriptional regulation network affect the transcription of overlapping sets of genes. Based on NET-seq data, we assessed the role of each factor in regulating nascent transcription, a more direct measurement of transcriptional phenotype than can be obtained from RNA-seq data. Nascent transcripts are produced antisense to the coding strand at substantial levels (*Churchman and Weissman, 2011*), so to obtain a complete and accurate view of expression differences, we annotated the antisense version of all genes and included these in differential expression analysis with DESeq2 (*Love et al., 2014*). First, we focused on sense protein-coding genes and inspected how many were differentially expressed across the strains. Interestingly, in some strains (e.g. *rph1Δ* and *nap1Δ*), very few genes were transcribed at significantly altered levels relative to the wild-type, whereas upon loss of Rpb4, a subunit of RNA polymerase, over 10% of all protein-coding genes were differentially transcribed (*Figure 1B*, *Figure 1—figure supplement 1B*; *Supplementary file 2*).

We then investigated the degree to which differentially transcribed genes were shared across mutant strains. Over 90% of differentially transcribed genes were identified in fewer than nine deletion strains, and 41% were differentially transcribed in only a single strain (*Figure 1C*). Only a few genes had altered expression in most of the deletion strains; some of these, such as *HSP12* are involved in stress responses, and their regulation may represent the cellular reaction to losing key transcription regulators.

We next asked whether certain biological functions or pathways were commonly affected across the deletion strains using GO enrichment analysis (*Figure 1D*, *Figure 1—figure supplement 1C*; *Supplementary file 3 Anders and Huber, 2010*; *Ashburner et al., 2000*; *Mi et al., 2019*; *The Gene Ontology Consortium, 2019*). Over 90% of GO pathways enriched among the differentially transcribed genes were identified in fewer than five deletion strains, with 56% identified in a single strain, emphasizing the largely distinct responses to loss of each factor (*Figure 1—figure supplement 1D*). GO enrichments were not particularly strong or specific overall (*Supplementary file 3*); however, we did detect enrichment of some pathways consistent with the known functions of certain factors. Upon deletion of *HPC2*, which encodes a subunit of the HIR nucleosome assembly complex involved in the regulation of histone gene transcription (*Formosa et al., 2002*; *Prochasson et al., 2005*; *Xu et al., 1992*), the term 'DNA replication-independent chromatin organization' (GO:0034724) was significantly enriched (100-fold enrichment, p-adjusted=0.021) among downregulated genes (*Figure 1D*). The GO term 'chromatin assembly factor (CAF-1) complex' (GO: 0033186) was enriched among downregulated genes only upon deletion of *CAC1*, *CAC2*, and *CAC3*, which encode CAF-1 subunits. Some

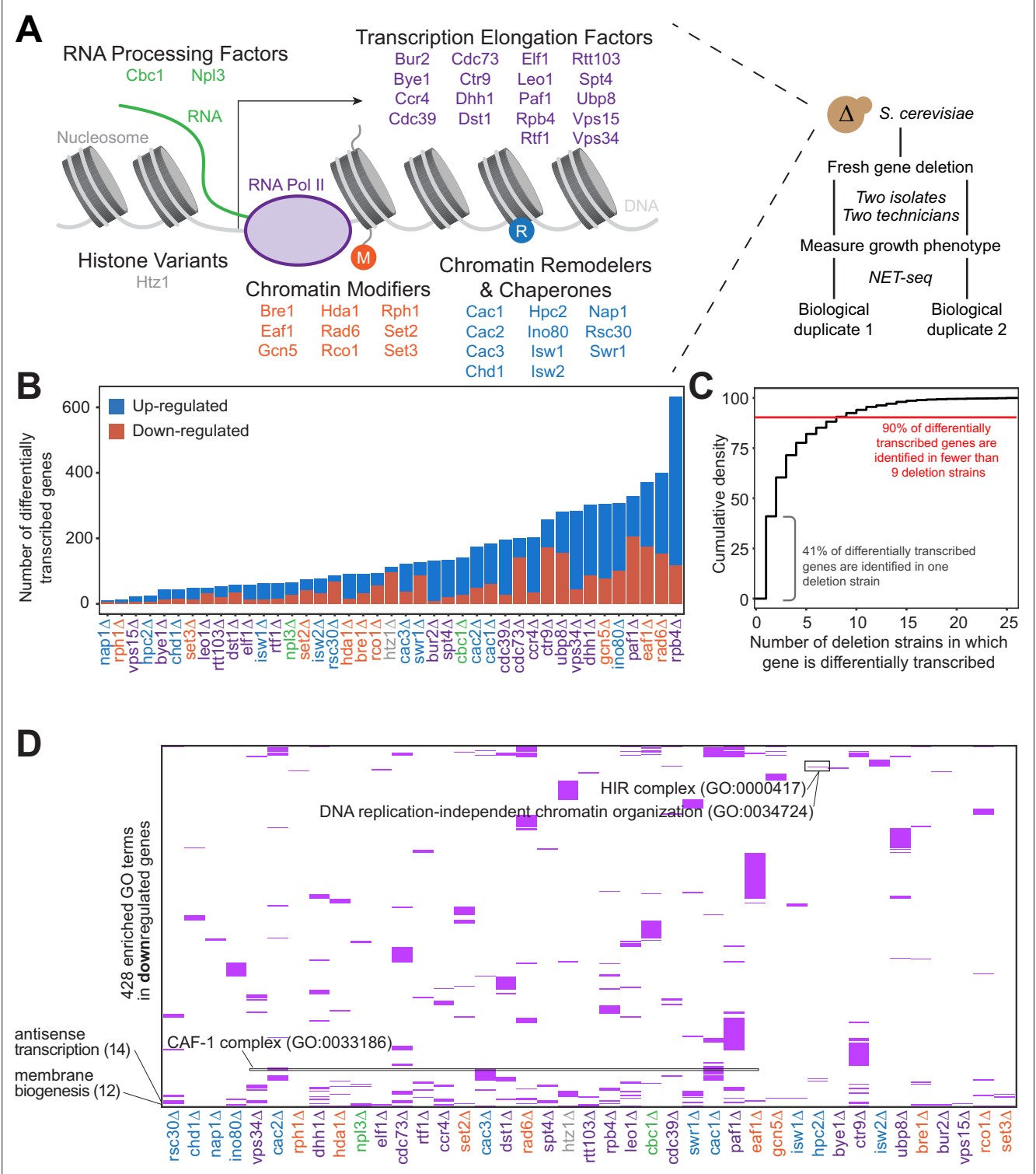

**Figure 1.** Gene expression is affected differently when transcription regulatory proteins are knocked out, both at the level of individual genes and gene ontology. (**A**) As polymerase II transcribes along a chromatinized template, a complex network regulates eukaryotic transcription elongation. Factors analyzed in the reverse genetic screen are listed and grouped by function: RNA processing factors (*green*), transcription elongation factors (*purple*), histone variants (*gray*), chromatin modifiers (*orange*), and chromatin remodelers and chaperones (*blue*). Colors of factors consistent throughout figures. Each of these factors were deleted to conduct a reverse genetic screen in *Saccharomyces cerevisiae*. For each deletion strain, a fresh gene deletion was conducted in two isolates by two technicians. After a growth phenotype was measured, native elongating transcript sequencing (NET-seq) was performed in at least two biological replicates. (**B**) A number of differentially up- (*blue*) and downregulated (*red*) genes vary widely across deletion

*Figure 1 continued on next page*

*Figure 1 continued*

strains. For differential expression analysis, all reads mapping to protein coding regions and their antisense counterparts were considered. Here, only sense genes are included in the counts. (**C**) Cumulative density plot illustrating that 41% of differentially expressed (DE) genes are only differentially transcribed in one strain, with 90% of DE genes differentially transcribed in nine strains or fewer. (**D**) A total of 420 gene ontology (GO) terms are enriched (*purple*) among the downregulated genes in at least one deletion strain; if a GO term is not enriched in a deletion strain's downregulated genes, the heatmap tile is white. Both axes are hierarchically clustered to group those deletion strains that share enriched ontologies. Numbers in parentheses to left of plot show the number of strains in which the GO term is enriched.

The online version of this article includes the following figure supplement(s) for figure 1:

**Figure supplement 1.** Native elongating transcript sequencing (NET-seq) screen data identifies largely different groups of genes with varying functions that are differentially expressed across deletion strains.

common functions were also revealed by GO analysis. The GO term 'Cvt complex' (GO: 0034270), a complex involved in autophagy, was enriched among upregulated genes in 17 deletion strains (*Figure 1—figure supplement 1C*), and the GO term 'membrane biogenesis' (GO: 0101025), was enriched among downregulated genes in 12 deletion strains (*Figure 1D*). These trends are consistent with the upregulation of autophagy and the downregulation of growth when transcription factors are deleted.

## Antisense transcription is misregulated upon deletion of transcription regulatory factors

NET-seq is uniquely suited to detect antisense transcription (*Figure 2A–B*), and since we included antisense transcripts in differential expression analysis, we have a direct readout of their expression. We linked every antisense transcript to the GO term 'antisense transcription' (GO: 9999999). This GO term was significantly enriched among downregulated genes in 14 strains and among upregulated genes in 6 strains (*Figure 1D*, *Figure 1—figure supplement 1C*).

To determine the effects of removing transcriptional regulators on antisense transcription, we visualized the spread of $\log_2$-fold changes vs wild-type of these antisense transcripts (*Figure 2C*). Our data revealed a continuum of median antisense transcription, with that of the wild-type strain near the middle of the range. The strains in which we observed the largest decrease in antisense transcription were those lacking factors relating to transcription elongation, such as Elf1, Rtt103, and the Pol II subunit Rpb4, suggesting an asymmetry in the impact of elongation factors on sense and antisense transcripts. The factors whose deletions led to the largest increase in the antisense transcription were those involved in the regulation of histone acetylation, including members of the Rpd3S–Set2 pathway (Set2) and the major histone H4 acetyltransferase complex NuA4 (Eaf1), emphasizing the role of acetylation/deacetylation in antisense transcription (*Carrozza et al., 2005*; *Churchman and Weissman, 2011*; *Krogan et al., 2003*; *Murray et al., 2015*; *Murray and Mellor, 2016*).

In many strains, changes in antisense transcription occurred in specific locations (*Figure 2—figure supplement 1A*). For example, increases in antisense transcription in the *dst1Δ* strain occurred primarily at the 3' end; in the *set2Δ* strain, antisense transcription increased uniformly across the gene; and in the *eaf1Δ* strain, antisense transcription increased within the gene, but not at the 3' end (*Figure 2D–F*). These findings imply that antisense transcription is a combination of different transcriptional activities regulated by separate sets of factors. Many of these factors had been identified as regulators of antisense transcription using northern blot analysis, microarrays, or other strategies (*Carrozza et al., 2005*; *Li et al., 2009*). In these cases, NET-seq analysis provides a higher-resolution picture that confirms and complements these earlier findings.

Antisense transcription can repress or activate sense transcription through direct (transcriptional interference) or indirect mechanisms, such as altered chromatin states (*Houseley et al., 2008*; *Lenstra et al., 2015*; *Martens et al., 2004*; *Nevers et al., 2018*; *Uhler et al., 2007*). However, it remains unclear whether changes in transcriptional output are generally connected to changes in antisense transcription across regulatory contexts (*Murray and Mellor, 2016*). We compared changes in gene transcription to changes in Pol II antisense transcription across a range of transcription regulatory landscapes. We found no correlation between antisense and sense transcriptional outputs when considering all strains together (*Figure 2—figure supplement 1B*). To determine whether any factor acts as a link between antisense and sense transcription, we plotted all Pearson r values across each strain individually (*Figure 2—figure supplement 1C*). Values ranged from –0.22 to 0.36, suggesting

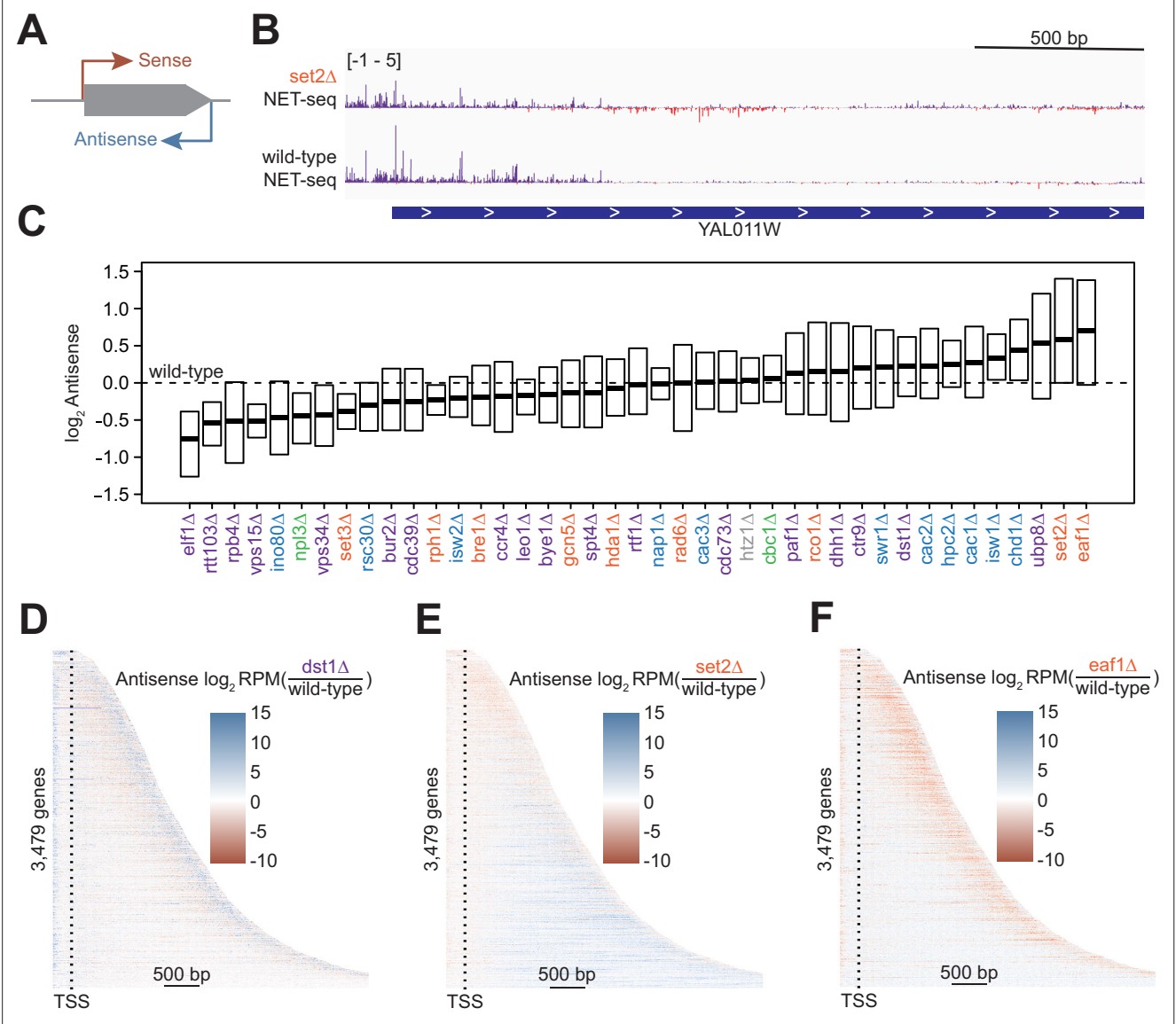

**Figure 2.** Antisense transcription is altered in most deletion strains. (**A**) Cartoon illustrating sense and antisense transcription of an example gene on the positive strand. (**B**) Wild-type and *set2Δ* native elongating transcript sequencing (NET-seq) data at YAL011W. Sense and antisense are displayed in purple and red, respectively. (**C**) Fold change in antisense transcription for each deletion strain compared to wild-type reveals that some strains have dramatically increased antisense transcription while others have much less than wild-type. Whiskers and outliers are omitted from visualization. (**D**) Heatmap of fold change in antisense transcription in the *dst1Δ* strain compared to wild-type reveals that most antisense transcription in the *dst1Δ* strain originates from the 3' end of genes. (**E–F**) Same as in (**D**), for *set2Δ* and *eaf1Δ*, respectively.

The online version of this article includes the following figure supplement(s) for figure 2:

**Figure supplement 1.** Antisense transcription is largely uncorrelated with gene length and not uniformly distributed across gene bodies.

that antisense transcription levels do not generally affect sense transcription in any of the regulatory contexts that we analyzed.

## Peaks of Pol II density across the gene body are altered in the absence of key transcription regulators

We found that Pol II density increases at loci critical for gene regulation, namely the transcription start sites (TSS), poly(A) sites, and splice sites (*Figure 3—figure supplement 1A-D*). At the 5' ends of genes, loss of Dst1, a homolog of the general transcription elongation factor TFIIS, dramatically increased Pol II pausing just downstream of the TSS (*Figure 3A*). We also observed peaks in Pol II

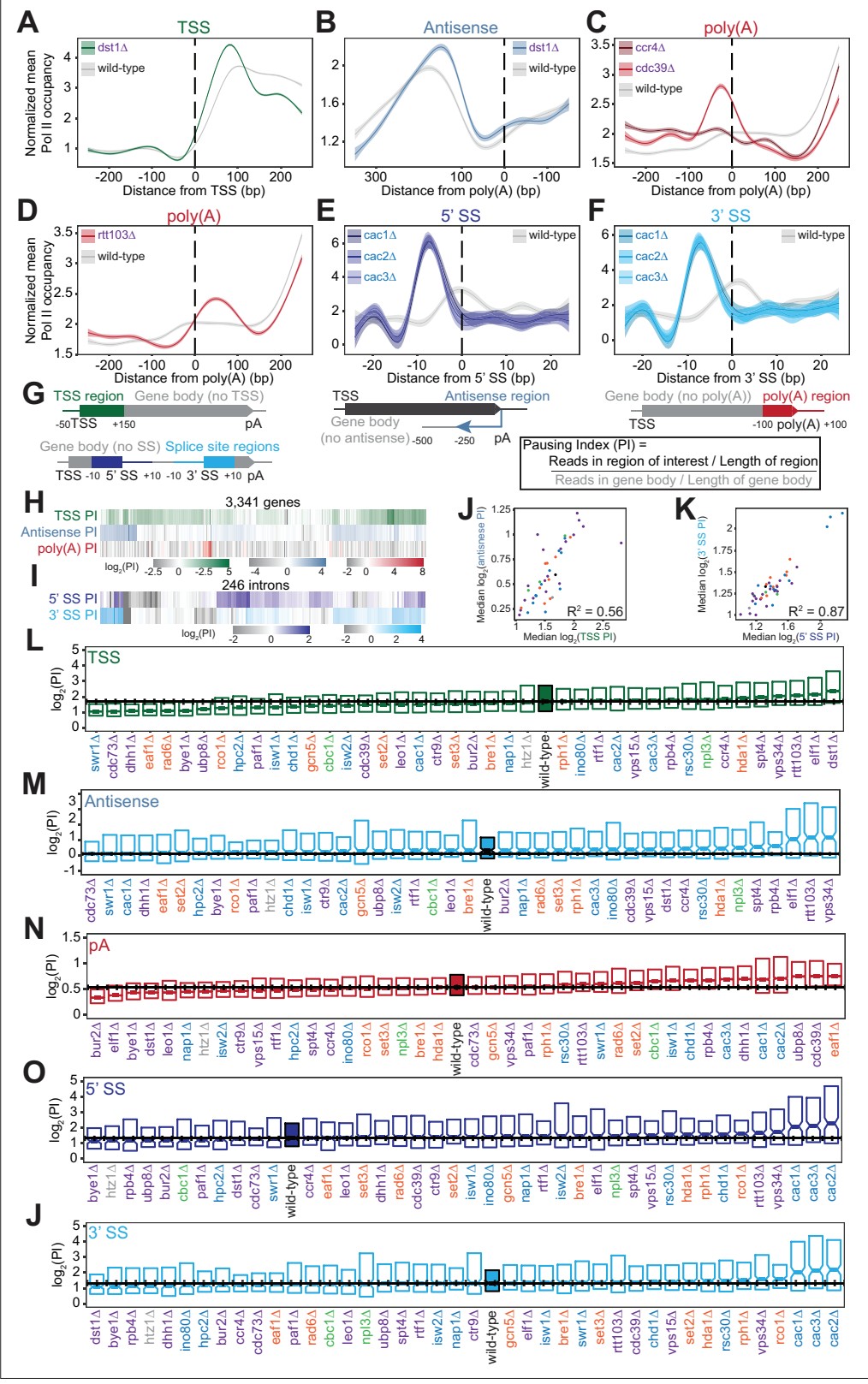

**Figure 3.** Polymerase II (Pol II) density is increased around transcription start sites (TSS), polyadenylation sites, and splice sites (SS). (**A**) Metagene plot of normalized mean Pol II occupancy and the surrounding 95% confidence interval for the 500 bp surrounding the most abundant annotated TSS (*Pelechano et al., 2013*) (n=2415 genes). Metagene for *dst1Δ* (*green*) can be compared to the Pol II density in the wild-type strain (*gray*). (**B**) Normalized

*Figure 3 continued on next page*

*Figure 3 continued*

mean Pol II occupancy and the surrounding 95% confidence interval for the 600 bp surrounding the most abundant annotated poly(A) sites (*Pelechano et al., 2013*) in the antisense orientation. Metagene for *dst1Δ* (*blue*) can be compared to the Pol II density in the wild-type strain (*gray*). (**C**) Normalized mean Pol II occupancy and the surrounding 95% confidence interval for the 500 bp surrounding the most abundant annotated poly(A) sites (*Pelechano et al., 2013*). Metagenes for subunits of the Ccr4-NOT complex deleted (*red*) can be compared to the Pol II density in the wild-type strain (*gray*). (**D**) Same as (**C**), for *rtt103Δ*. (**E–F**) Normalized mean Pol II occupancy and the surrounding 95% confidence interval for the 50 bp surrounding annotated 5′ and 3′ splice sites (SS). Metagenes for subunits of the Caf1 complex deleted (*blue*) can be compared to the Pol II density in the wild-type strain (*gray*). (**G**) Cartoon and equation illustrating pausing index (PI) calculation. (**H**) PI for the TSS (*green*), polyadenylation [poly(A)] (*red*), and 3′ antisense (*blue*) regions across genes. Horizontal axis is hierarchically clustered, revealing TSS, poly(A), and antisense pausing indices for genes in wild-type yeast. (**I**) Same as (**H**), for 5′ and 3′ SS pausing indices. (**J**) Scatter plot of the median pausing indices in the TSS and 3′ antisense regions for all deletion strains. Relationship was quantified using Pearson correlation. (**K**) Same as in (**J**), comparing pausing the 5′ and 3′ SS surrounding introns. (**L**) Boxplot of TSS PI distributions in each deletion strain, ordered by median PI. Horizontal solid line indicates median value for wild-type yeast; dotted lines indicate the 45th and 55th percentile of wild-type PI values. (**M–P**) Same as (**L**), for 3′ antisense PI, poly(A) site PI, 5′ SS PI, and 3′ SS PI.

The online version of this article includes the following figure supplement(s) for figure 3:

**Figure supplement 1.** Heatmaps of polymerase II (Pol II) density around RNA processing sites reveal differences in polymerase behavior across deletion strains, which can have functional consequences in specific deletion strains.

**Figure supplement 2.** Polymerase II density is increased around RNA processing sites to varying degrees across deletion strains.

---

density at the start of antisense transcripts opposite the 3′ ends of genes. Interestingly, deletion of *DST1* had an effect on antisense transcription similar to its impact on sense transcription (*Figure 3B*).

At the 3′ ends of genes, we observed changes in Pol II density upon loss of factors that regulate termination or polyadenylation. The screen included two subunits of the Ccr4-Not complex, which plays many roles in gene regulation including deadenylation (*Figure 3C*; *Funakoshi et al., 2007*; *Raisch et al., 2019*; *Temme et al., 2014*; *Tucker et al., 2002*; *Wahle and Winkler, 2013*; *Yamashita et al., 2005*; *Yi et al., 2018*). Deletion of the scaffolding Cdc39 subunit of the complex resulted in substantial pausing before poly(A) sites, followed by reduced Pol II density. By contrast, loss of the catalytic Ccr4 subunit decreased density only downstream, with a much less prominent upstream pause (*Figure 3C*). Loss of proteins more directly involved in transcription termination, such as Rtt103, resulted in Pol II stalling just downstream of poly(A) sites, suggesting that Pol II may slow down during recruitment of this termination factor (*Figure 3D*). In these deletion strains and others, the locations of 3′-end Pol II peaks varied, with some strains exhibiting a Pol II peak before poly(A) sites and others exhibiting a peak after (*Figure 3—figure supplement 1C*) indicating that Pol II is controlled both before and after poly(A) sites.

Pol II density increases around splice sites upon the loss of several transcription regulators. For example, pause indices increased most strongly when any of the CAF-I complex components (i.e. Cac1, Cac2, Cac3) were deleted (*Figure 3E–F*). CAF-I promotes histone deposition onto newly synthesized DNA (*Kaufman et al., 1997*), and to the best of our knowledge has not been implicated in splicing. To determine whether splicing is altered upon loss of CAF-1, we analyzed *cac2Δ* RNA-seq data (*Hewawasam et al., 2018*). We detected a modest but statistically significant increase in splicing in the *cac2Δ* strain relative to the wild-type (p=0.02; *Figure 3—figure supplement 1E, F*). Thus, CAF-1 decreases Pol II density at splice sites and regulates splicing, suggesting that the complex links Pol II pausing with splicing efficiency.

To quantify Pol II pausing at each site, we defined a pausing index (PI), a length-normalized metric comparing Pol II density in the region of interest to that in the rest of the gene body (*Figure 3G*). Interestingly, genes with a high PI in one location did not tend to have a high index for other locations (*Figure 3H–I*). Overall, at the per gene level, there was a poor correlation between all pausing indices in the wild-type strain (e.g. TSS PI vs poly(A) PI for each gene has $R^2$=0.06; all $R^2$ ≤0.10, p>0.05; *Figure 3—figure supplement 2A*). Even across each intron, pause indices differ at 5′- and 3′-splice sites although strong pausing occurs at 5′ splice sites as often as at 3′ splice sites (*Figure 3—figure*

*supplement 2B*). Thus, pause indices vary across each gene, from the TSS to poly(A) sites, suggesting that each region of high Pol II density is regulated in a different manner.

Across deletion strains, the median PI varied, with the wild-type indices lying near the middle of the dynamic range (*Figure 3L–P*, S4D-H). For example, the median TSS PI ranged from 1.06 in *cdc73Δ* to 2.81 in *dst1Δ,* with wild-type at 1.68 (*Figure 3L*, S3A). The levels of antisense pausing also vary substantially across the strains (*Figure 3M*).

We asked whether the same factors are implicated in regulating the different Pol II peaks. Indeed, there was a relatively strong correlation between median TSS pausing indices and antisense pausing indices across the deletion strains ($R^2$=0.56, p<0.001; *Figure 3J*). Of the 10 strains with the highest TSS pausing indices, 9 were also in the top 10 for median antisense pausing indices (*Figure 3L–M*). These strains tended to lack known elongation factors, such as Dst1 and Spt4, indicating the role of transcription elongation factors in relieving pausing at the start of transcription. In addition, factors that modulate pausing at splice sites tended to do so at both sites overall, but not at the same intron ($R^2$=0.87, p<0.001; *Figure 3K*, S4B). However, we did not observe similar relationships between other pause indices (*Figure 3—figure supplement 2C*). For example, factors impacting pausing near the TSS do not have a similar impact at splice sites or at poly(A) sites, indicating that different mechanisms control Pol II pausing in different genic regions.

## Pol II pausing locations are affected by deletion of transcription regulators

Along with identifying regions of elevated Pol II density, NET-seq data pinpoints precise positions that Pol II pauses within regions of high Pol II density and elsewhere. Because NET-seq is performed in bulk on a population of cells, only the sites that consistently induce pausing are observed, and we refer to these as 'stereotypical' pause positions. These precise sites of Pol II pausing at single nucleotides are reminiscent of in vitro RNA polymerase pausing observed at specific positions of DNA templates (*Galburt et al., 2007*; *Hodges et al., 2009*; *Kingston and Chamberlin, 1981*; *Mayer et al., 2017*; *Wang et al., 1998*). We systematically identified putative pause sites in strains with sufficient coverage as positions with read densities that deviate from the statistical fluctuations of the surrounding 200 nucleotides, modeled as a negative binomial distribution (>3 standard deviations from the mean; *Figure 4A–B*). Using an irreproducibility discovery rate (IDR) analysis, the putative pause sites are ranked and compared across replicates (*Landt et al., 2012*; *Li et al., 2011*). Pause sites that correspond across replicates using an IDR threshold of 1% are considered reproducible and used for downstream analyses. Approximately, one-third of the initially called pause sites is determined reproducible between two wild-type replicates using this criteria, but the majority of reproducible pause sites using various combinations of replicates overlap (*Figure 4—figure supplement 1A, B*). Stereotypical pause sites in NET-seq data represent loci where Pol II pauses in many cells and represent a fraction of the overall pausing by Pol II. The *E. coli* RNA polymerase pauses both at specific pause sites and randomly across a DNA template (*Adelman et al., 2002*; *Neuman et al., 2003*). Thus, Pol II is likely to similarly pause ubiquitously across gene bodies in noncanonical ways that would not lead to a detectable signal in NET-seq data. Nevertheless, the stereotypical pause sites identified here provide insight into the underlying features that induce Pol II pausing.

In NET-seq analysis and other 3' end mapping approaches, mispriming events during reverse transcription (RT) can occur when the RT primer anneals internally within the nascent RNA rather than with the oligo ligated to the 3' end (*Gajos et al., 2021*; *Mayer et al., 2015*). RT mispriming is far more likely to occur on nascent RNA derived from large genomes as there are many more sequences that could be recognized by the RT primer. Such events can be identified computationally and removed as the reads lack a unique molecular identifier sequence and align proximal to sites complementary to the RT primer. To reduce their occurrence in the first place a nested NET-seq library strategy has been employed to lessen mispriming in human NET-seq analysis (*Gajos et al., 2021*). In yeast, we found that the nested NET-seq library approach does not change the number of pauses identified (*Figure 4—figure supplement 1C*) nor does it decrease the fraction of pause sites with adapter-like sequence downstream, which is expected at sites of mispriming (*Figure 4—figure supplement 1D*). We similarly found that the number of pauses identified with and without removing reads with identical molecular barcodes ('PCR duplicates') shows virtually the same number of pause sites (*Figure 4—figure supplement 1E*). Before identifying the locations of pause sites, we computationally removed all reads that

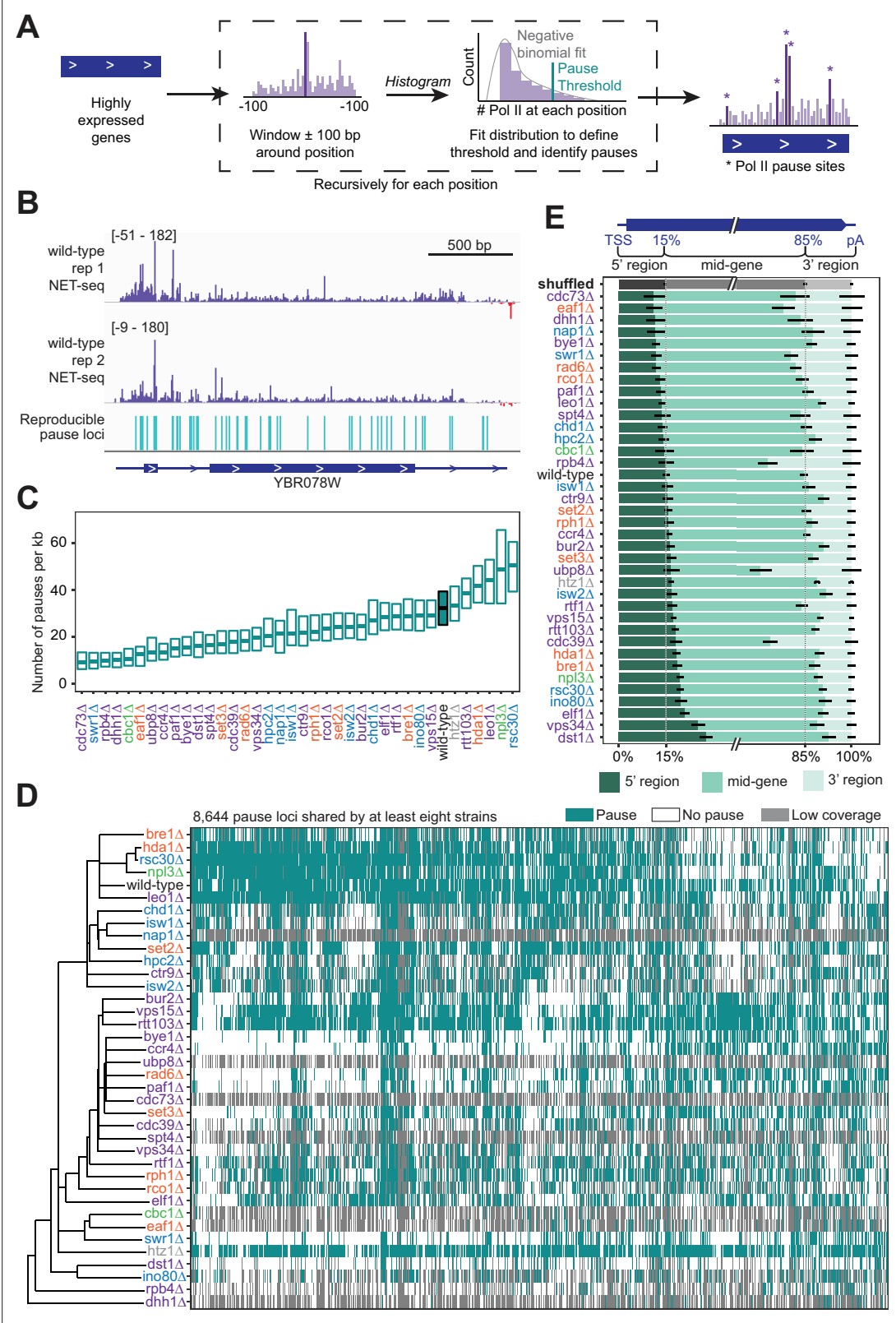

**Figure 4.** Trends in polymerase II (Pol II) pausing behavior at single-nucleotide resolution across deletion strains. (**A**) Cartoon illustrating algorithm for robust and reproducible Pol II pause detection. (**B**) Example of Pol II density on the positive (*purple*) and negative (*red*) strands, as measured by native elongating transcript sequencing (NET-seq) in two wild-type replicates. Pauses that meet the 1% irreproducibility discovery rate (IDR) reproducibility threshold are shown as blue vertical lines. (**C**) Boxplot of the distribution of Pol II pause densities, the number of pauses per kilobase examined, in

*Figure 4 continued on next page*

*Figure 4 continued*

each deletion strain, ordered by median pausing density. Whiskers and outliers were removed for visualization. (**D**) Hierarchically clustered heatmap of 8644 Pol II pause loci across the genome reveals locations of pauses shared by multiple deletion strains. Heatmap is colored based on if that locus was identified as a pause (*teal*), not a pause (*white*), or if there was not sufficient coverage to determine pause status (*gray*). Analyses conducted only on deletion strains with biological replicates and only at loci at which there was enough coverage to determine the absence of a Pol II pause in at least one deletion strain. (**E**) The percent of Pol II pause loci located in the 5′ gene region, mid-gene, and 3′ gene region varies across deletion strains. The 5′ gene region was identified for each well-expressed gene as extending from the transcription start site to the 15th percentile of the gene length. Similarly, the 3′ gene region was defined as the last 15th percentile of the gene length, with the mid-gene region spanning in between. The control (*gray*) was created by scrambling all identified pauses across all deletion strains within the genes they were identified in. Rows are ordered by the percent of pauses found in the 5′ region. Bars represent the 95% confidence intervals across all expressed genes.

The online version of this article includes the following figure supplement(s) for figure 4:

**Figure supplement 1.** Polymerase II (Pol II) pausing behavior at single-nucleotide resolution across deletion strains reveals that pausing is balanced and dynamic in wild-type.

are due to RT mispriming, but to avoid possible distortions that occur during deduplication, we did not remove putative PCR duplicates (*Fu et al., 2018*; *Parekh et al., 2016*).

We calculated the pause site density, or the number of sites per kilobase, for genes that had sufficient coverage. The density varied widely across deletion strains (*Figure 4C*), which cannot be explained by differences in sequencing depth across deletion strains ($R^2$=0.003, p=0.743; *Figure 4—figure supplement 1F*). In the wild-type strain, we found Pol II pause sites every 33 bp on average. Some of the deletion strains exhibited more pausing overall at stereotypical pause sites; for example, upon loss of Rsc30, a subunit of the RSC chromatin remodeling complex, 33% of all NET-seq reads mapping to highly expressed genes constituted pause sites, versus only 21% in the wild-type (*Figure 4—figure supplement 1G*). Thus, the RSC complex obscures Pol II pause sites, which is likely related to its role in diminishing the nucleosomal barrier to Pol II elongation (*Carey et al., 2006*). Perhaps unexpectedly, loss of canonical transcription elongation factors, such as Spt4 and Dst1, resulted in a lower pause site density relative to the wild-type (*Figure 4C*). However, pause site density describes only one feature of Pol II elongation. The density includes only the stereotypical locations at which Pol II typically pauses in many cells, so it is not a measure of the absolute frequency of Pol II pausing. In addition, the densitiy is not related to the Pol II catalysis rate. Thus, these transcription elongation factors may facilitate other aspects of transcription elongation or they may act locally to influence Pol II during specific points of regulation, consistent with their impact on peaks of Pol II density only near TSS (*Figure 3L*).

The pause loci for each strain included many that were not observed in wild-type yeast (*Figure 4D*). Indeed, when the sets of pause loci are used to cluster deletion strains by principal component analysis, the wild-type strain stands away from most strains (*Figure 4—figure supplement 1H*). However, some deletion strains shared many pause sites with those observed under in the wild-type: 81% of pause sites identified in wild-type yeast were also identified in the *htz1Δ* strain, consistent with its confined role at the +1 nucleosome (*Bagchi et al., 2020*; *Zhang et al., 2005*).

We wondered whether loss of related factors would lead to the same sets of pause sites. We first identified all pause sites observed in at least eight strains and used the presence or absence of these pauses in each strain to perform hierarchical clustering (*Figure 4D*). *dst1Δ* pause sites clustered far away from those in wild-type cells, consistent with the backtracking role of Dst1 that leads to downstream-shifted pause sites (*Churchman and Weissman, 2011*; *Noe Gonzalez et al., 2021*). H2B ubiquitination increases the nucleosomal barrier to Pol II (*Chen et al., 2019*), so alterations to histone ubiquitination might lead to new pause sites. Interestingly, pause sites after the loss of Rad6, Ubp8, Paf1, and Cdc73 all cluster together. Rad6 and Ubp8 ubiquitinate and deubiquitinate H2B, respectively (*Amerik et al., 2000*; *Jentsch et al., 1987*). Paf1 and Cdc73, members of the Paf1 complex, are responsible for recruiting Rad6 to chromatin (*Kim and Roeder, 2009*). The clustering of these factors indicates a role for H2B ubiquitination in determining the locations of many pause sites. Finally, we figured that differences in nucleosome positioning may lead to differential pause sites usage, so we inspected how pause sites change after the loss of different chromatin remodelers. Interestingly, we observed that loss of ISWI and CHD chromatin remodelers, Isw1, Isw2, and Chd1, leads to pause sites that cluster together (*Figure 4D*). For example, most of the pause sites observed in *isw1Δ* (76%) were also observed in *chd1Δ*, consistent with their joint roles in maintaining chromatin structure (*Ocampo et al., 2016*; *Smolle et al., 2012*). In contrast, loss of INO80, SWR1, and SWI/SNF family remodelers,

Ino80, Rsc30, and Swr1, all leads to distinct sets of pause sites consistent with their separate roles in chromatin remodeling (*Figure 4D*; *Singh and Mueller-Planitz, 2021*).

Pol II pause sites in the wild-type strain were distributed evenly throughout gene bodies (*Figure 4E*). By contrast, deletion strains exhibited a range between twofold decreased and twofold increased Pol II pause sites in the 3' regions of genes, with slightly less variability at the 5' regions of genes relative to a scrambled control or wild-type pausing (*Figure 4E*). The enrichment of pause sites at 5' end and 3' regions generally corresponds with our PI results (*Figure 3H, L and N*). For example, deletion of *DST1* approximately doubled pause loci in the 5' regions at the expense of pausing in 3' regions. This localized effect exemplifies how overall pause density (see *Figure 4C*) of a gene could be decreased in a deletion strain lacking a canonical elongation factor. However, in general, changes in 5' vs 3' pause sites in deletion strains were not correlated (*Figure 4E*). We find substantially more pause sites at the 3' regions of genes in *rpb4Δ*. Rpb4 is a Pol II subunit that dissociates with the complex at the ends of genes (*Mosley et al., 2013*) and is responsible for sustained transcription elongation through the 3' ends of genes (*Runner et al., 2008*). Thus, Rpb4 prevents Pol II from pausing at the 3' regions of genes that may protect from premature termination before the canonical 3' cleavage site is transcribed. Similarly, more 3' pause sites are found in the *ubp8Δ* strain, consistent with the global increase in this strain of H2B ubiquitination, a mark that increases the nucleosomal barrier to Pol II and is coincident with Pol II pausing at transcription termination sites (*Bonnet et al., 2014*; *Chen et al., 2019*; *Harlen et al., 2016*). Together, these data show how the chromatin landscape and transcriptional regulatory network of the cell dictate stereotypical sites of Pol II pausing that in turn controls where and for how long Pol II pauses during elongation.

## Chromatin and DNA features can accurately classify Pol II pausing locations in deletions strains

Given the number of reproducible pause sites we identified, we next investigated whether we could determine which genomic features, if any, were responsible for the stereotypical pause sites. In vitro studies have shown that Pol II pausing has many causes, including specific DNA sequences, nucleosomes, and histone modifications (*Bintu et al., 2012*; *Herbert et al., 2006*; *Hodges et al., 2009*; *Kassavetis and Chamberlin, 1981*; *Kireeva et al., 2005*; *Kireeva and Kashlev, 2009*; *Shaevitz et al., 2003*). In vivo, the dominant factors globally associated with Pol II pause sites remain unclear, although sequence elements, transcription factors, nucleosomes, and CTD modifications have all been connected to Pol II pausing (*Alexander et al., 2010*; *Churchman and Weissman, 2011*; *Gajos et al., 2021*; *Nechaev et al., 2010*; *Noe Gonzalez et al., 2021*; *Nojima et al., 2018*; *Shukla et al., 2011*). Recently, DNA sequence and shape were shown to be important contributors to pause site locations in human cells (*Gajos et al., 2021*). We first asked whether specific DNA sequences were connected with Pol II pausing loci. Previous studies reported that Pol II has a strong bias toward pausing at adenine (*Churchman and Weissman, 2011*), which we also observed here. More specifically, we observed a 3.4-fold enrichment of real Pol II pause sites at TAT trinucleotide sequences relative to shuffled control sites in the same well-expressed genes (*Figure 5A*). The shape of the DNA itself, as predicted from sequence, also appears to inform the location and propensity for Pol II to stall: DNA low helix twist values were more common under real pause loci than in the shuffled control (*Figure 5B*). These observations were consistent, as the AT dinucleotide step has a low average twist angle of 32.1° (*Ussery, 2002*).

Beyond the trinucleotides, significantly enriched sequence motifs were also associated with Pol II pause sites in most deletion strains (*Supplementary file 4*), including three motifs related to pauses in the wild-type strain (*Figure 5C*). Notably, not all motifs are shared across strains, and upon deletion of some factors, new motifs were associated with Pol II pause sites. 13 of the 26 identified sequence motifs with high relative entropies significantly matched known transcription factor binding site motifs (*Figure 5D*). Thus, it is likely that Pol II pause sites can partially, but not fully, be explained by DNA sequence and/or proteins binding to DNA.

In addition to the structure of the DNA itself, chromatin features, such as nucleosome positions and histone modifications, are also connected to Pol II pausing behavior. To search broadly for genomic features underlying sites of Pol II pausing, we evaluated 51 features (*Supplementary file 5*), including nucleotide sequence, DNA shape, position of pauses within a gene, histone modifications, and Pol II CTD phosphorylation marks. 35 out of 42 exhibited a statistically significant difference between

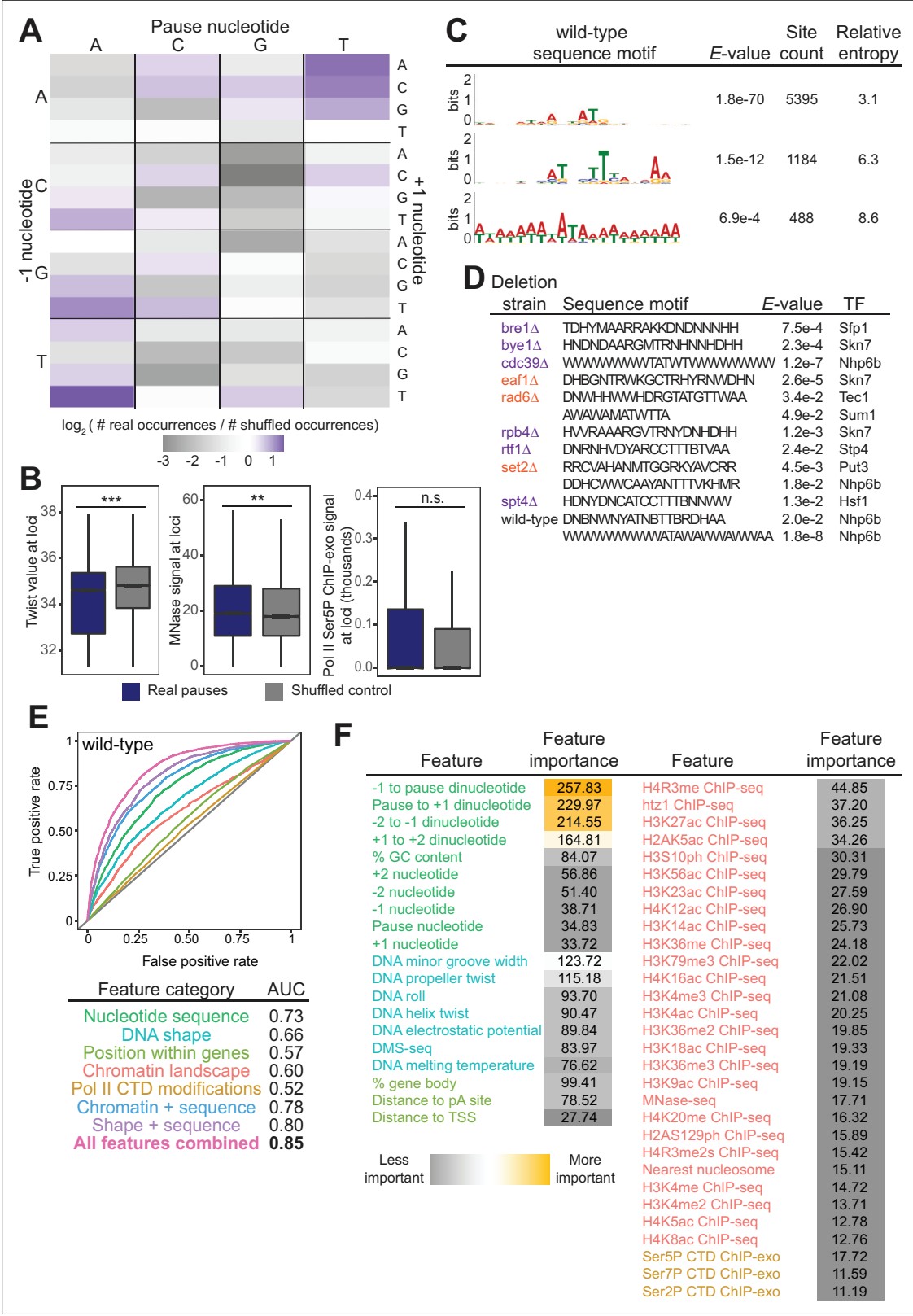

**Figure 5.** Chromatin and DNA features explain the location of some polymerase II (Pol II) pauses in wild-type yeast. (**A**) Heatmap illustrating the relative frequency of each trinucleotide sequence surrounding real and shuffled control pauses centered on Pol II pauses identified in wild-type. (**B**, left) Comparison in the distribution of values for twist values underlying Pol II pauses in wild-type yeast (n=13,994) compared to a shuffled control, in which the same number of pauses is shuffled, maintaining the same number of pauses within each well-expressed gene. Differences between the real

*Figure 5 continued on next page*

*Figure 5 continued*

and shuffled distributions were determined as significantly significant by a Student's t-test with Bonferroni correction for multiple hypotheses. p-values are reported in **Supplementary file 5**. (* adjusted p-value ≤0.05; ** adjusted p-value ≤0.01; *** adjusted p-value ≤0.001). Also shown for MNase-seq signal (center) and Ser5P CTD ChIP-exo signal (right). (**C**) Table showing the three significant motifs identified under Pol II pauses in the wild-type strain. All analyses were performed using the MEME suite of tools. Significant motifs were those with an *E*-value greater than 0.05. Pause sites were scrambled within well-expressed genes to be used as a negative control and to calculate enrichment of motifs. (**D**) Table with all sequence motifs underlying pauses across deletion strains that are significantly similar to known transcription factor binding motifs. Only the top match, as assessed by E-value, is reported. (**E**) Receiver operating characteristic curve from a random forest classifier that measures the predictive value of chromatin and DNA features on Pol II pauses in wild-type yeast (10,495 training and 3499 training loci). (**F**) Table of all features used in random forest classifier for pause loci classification and the importance of each feature. Feature importance is calculated as the mean decrease in accuracy upon removing that feature from the model.

The online version of this article includes the following figure supplement(s) for figure 5:

**Figure supplement 1.** Chromatin and DNA features explain the location of some polymerase II (Pol II) pauses in wild-type yeast using a random forest classifier.

real wild-type pause sites and shuffled controls (the remaining 9 of the 51 are sequence features that cannot be compared on a numeric scale) (***Figure 5—figure supplement 1A***, ***Supplementary file 6***). For example, the MNase-seq signal around pause loci and the distance to the nearest nucleosome differed significantly between real and shuffled pause sites (***Figure 5B***, S6A), consistent with observations of pauses at nucleosomes (***Churchman and Weissman, 2011***). Interestingly, Ser2, Ser5, and Ser7 phosphorylation of the Pol II CTD did not differ relative to random positions, indicating that connections between Pol II phosphorylation and pausing at intron-exon boundaries are specific to pausing at those loci (***Alexander et al., 2010***). Among the features that differed significantly was DNA melting temperature, which was previously shown to influence Pol II stalling (***Nechaev et al., 2010***).

To determine which features may underlie where Pol II pauses, we created a random forest classifier to discriminate between real and shuffled control Pol II pause sites based on the surrounding chromatin and DNA features. A random forest classifier using all 51 features performed well (AUC = 0.85, ***Figure 5E***) relative to a random model (AUC = 0.5) at classifying Pol II pauses in wild-type yeast. Which features contribute the most to the random forest classifier can help shape models for the molecular underpinnings of stereotypical Pol II pausing. The most critical features for accurate identification of Pol II pause sites were DNA sequence surrounding the pause locus and topology features of the DNA at that locus (***Figure 5F***). A reliance on DNA sequence and DNA shape for determining pause sites was also observed in human NET-seq data despite a different DNA motif (***Gajos et al., 2021***). Together, these analyses showed that DNA sequence and shape contribute strongly to Pol II pause locations, but their effects are enhanced by many other features.

To ask whether features underlying Pol II pausing vary in different regulatory and chromatin landscapes, we built random forest models for each deletion strain. Across all deletion strains, an AUC of at least 0.78 was attained. These AUC values were only partially correlated with the total number of pauses detected in each deletion strain ($R^2$=0.37, p=0.000064; ***Figure 6—figure supplement 1A***). Although nucleotide sequence and DNA shape were the most important features for classifying Pol II pause loci in the wild-type and many deletion strains, models for a subset of strains (including *cdc39Δ, dst1Δ, ubp8Δ*) revealed that wild-type chromatin modifications were more powerful for Pol II classification (***Figure 6A***, S7B-E). We next performed a transfer of learning analysis to ask how each model would perform when classifying pauses in other strains. When trained on Pol II pause sites identified in wild-type yeast, the AUC when testing on pauses across all other strains ranged from 0.53 (*cbc1Δ*) to 0.82 (*vps15Δ*), revealing the differences across the strains (***Figure 6B***). We previously observed that loss of Dst1 leads to ~75% of pause sites to shift downstream (***Churchman and Weissman, 2011***). Thus, training a model on *dst1Δ* pause sites should not do well to classify pauses in another strain. Indeed, a model trained on *dst1Δ* pause sites performed well in classifying *dst1Δ* pause sites (AUC = 0.83); however, it performed the worst of all models in classifying pause sites in other deletion strains, obtaining a median AUC of 0.63 across them. These models indicate that the nucleotide sequence, DNA topology, position within a gene, and chromatin landscape all play roles in determining the location of Pol II pauses during transcription elongation.

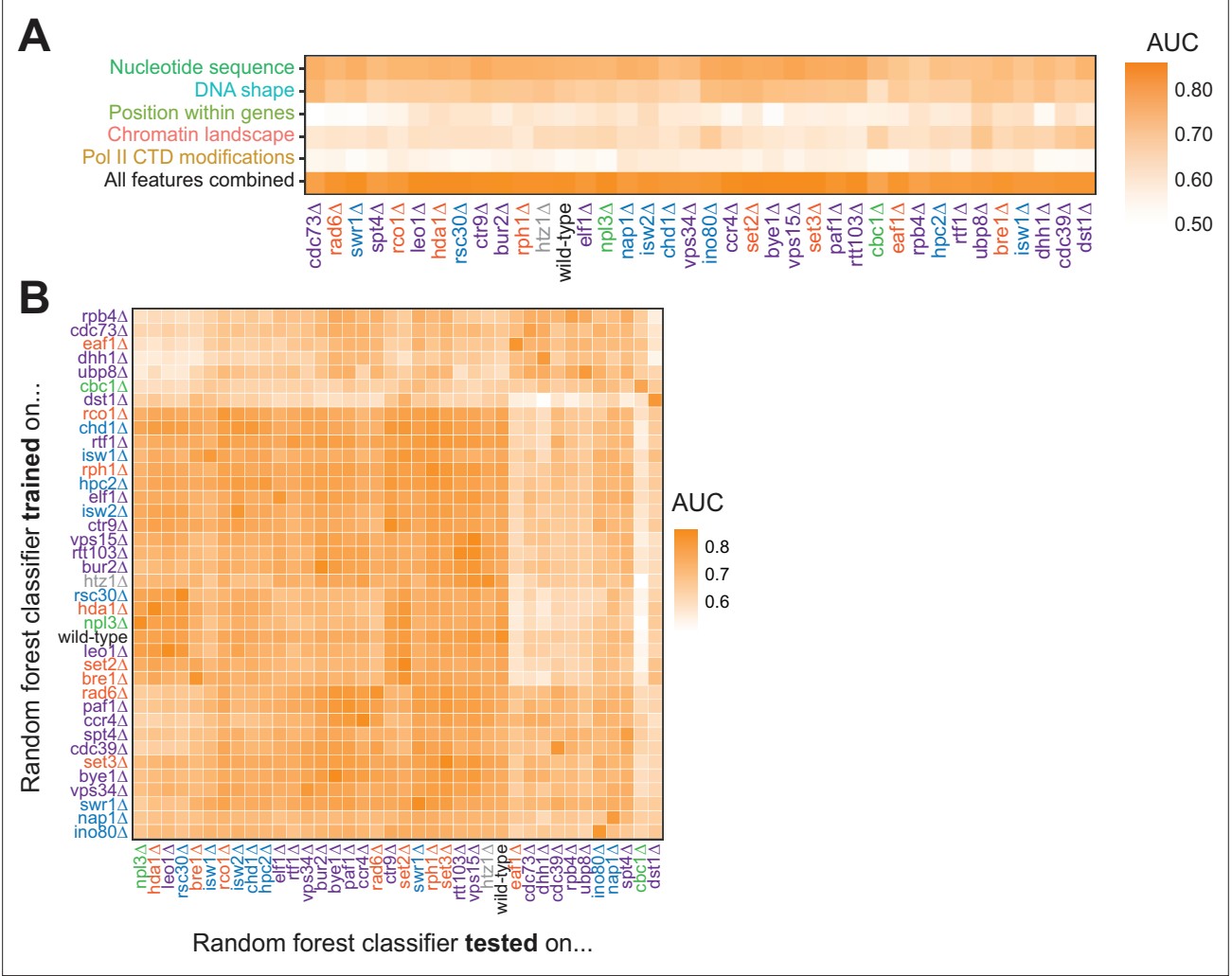

**Figure 6.** Random forest classifiers identify polymerase II (Pol II) pause loci across deletion strains, with different feature importance values across deletion strains. (**A**) Heatmap illustrating the mean AUC for the random forest classifier when trained (75% of loci) and tested (25% of loci) on each deletion strain. Deletion strains are hierarchically clustered along the x-axis. (**B**) Heatmap showing the AUC values from random forest classifiers trained on all pauses from one deletion strain (y-axis) and tested on those unique pauses observed in another deletion strain (x-axis). Both axes are hierarchically clustered to reveal similarities in AUC values across deletion strains. Tiles when the same training and testing strain are indicated are colored according to the AUC for that deletion strain when 75% of pauses in that deletion strain are used for training and the remaining 25% are used for testing as reported in (**A**).

The online version of this article includes the following figure supplement(s) for figure 6:

**Figure supplement 1.** Random forest classifiers can predict polymerase II pause loci across deletion strains, with different feature importance values across deletion strains.

## Discussion

Advances in high-throughput sequencing of nascent RNA have revealed that, in many eukaryotes, the vast majority of the genome is transcribed (*Hangauer et al., 2013*; *Struhl, 2007*). Nevertheless, this broad transcriptional activity is one of the most highly regulated processes within the cell. Multiple levels of regulation are orchestrated by DNA sequence, transcription factors, RNA processing factors, and chromatin modulators. Here, we used NET-seq to study 41 factors with connections to transcription elongation and discovered the remarkable tunability of transcription elongation. For all of the transcriptional phenotypes analyzed, the wild-type strain fell in the middle of the dynamic range observed across the deletion strains, revealing the intricate balance of transcriptional activity.

The 41 factors chosen for this study were previously annotated to regulate transcription elongation. However, loss of each factor had a unique impact on gene expression, suggesting that genes are

differentially sensitized to perturbations of the transcription regulatory network. Levels of antisense transcription in the deletion strains vary across a broad dynamic range, revealing that antisense transcription is finely tuned by many factors. Interestingly, loss of 20 factors decreased antisense transcription in cells (*Figure 2C*), indicating that it is possible to suppress antisense transcription further than what is observed in wild-type. Conversely, loss of 14 factors increased antisense transcription. Together, these results imply that wild-type antisense transcription is balanced by the influence of many factors and, in turn, can be precisely controlled. The possibility of tight control of antisense transcription indicates that regulatory mechanisms can exist where antisense transcription impacts sense transcription, consistent with the mechanisms described thus far (*Hongay et al., 2006*; *Houseley et al., 2008*; *Lenstra et al., 2015*; *Martens et al., 2004*; *Uhler et al., 2007*). Although, we did not observe a general correspondence between sense transcription and antisense transcription in this study.

Peaks of Pol II density were detected near TSSs, poly(A) sites, and both 5' and 3' splice sites. Interestingly, factors that impacted pausing at the 5' ends of genes were not the same as those that impacted pausing at 3' ends or at SS. Clearly, different mechanisms regulate Pol II pausing at different points during elongation. However, pausing around the TSS and pausing during antisense transcription were controlled by a similar set of factors that are highly enriched for established transcription elongation factors, such as *SPT4* and *DST1*. These findings suggest that there is a checkpoint early in transcription, in the sense and antisense directions.

Unexpectedly, we found that loss of the CAF-1 complex leads to pronounced Pol II peaks at 5' and 3' splice sites (*Figure 3E and F*). The CAF-1 complex is characterized as a chromatin assembly factor that promotes nucleosome assembly on newly synthesized DNA, sets the size of nucleosome depleted regions, and suppresses divergent transcription (*Fennessy and Owen-Hughes, 2016*; *Kaufman et al., 1997*; *Marquardt et al., 2014*). In addition, our findings connect the complex to splicing. It is tempting to speculate that loss of the CAF-1 complex leads to poorly deposited nucleosomes near SS, which alters Pol II pausing and co-transcriptional splicing.

Within the regions of elevated Pol II density (e.g. TSSs and SS) and across gene bodies are discrete pauses at single nucleotides that represent locations where Pol II has a higher propensity to pause. This set of positions varies substantially across the deletion strains (*Figure 4D*), indicating that there are a large number of possible pause sites, but the presence of regulatory factors modulates the pausing landscape such that they are not utilized. Our machine learning models of pause site preferences found that DNA sequence and shape are the most influential, followed by the chromatin landscape. We propose that the DNA template presents a varying energy landscape to the elongating Pol II through sequence variation and that nucleosome positions alter the landscape by lowering or enhancing pausing energetics and the associated chance of Pol II pausing. We also found that some transcription factor binding sites are enriched near pause sites, indicating a possible role for DNA binding proteins in Pol II pausing. A future analysis of the role of transcription factors, RNA binding proteins, and RNA structure in pausing would be an interesting avenue of investigation.

This work reveals the complex regulation of transcription elongation by a network of factors. In addition, it serves as a resource of NET-seq data to explore more specific hypothesis-driven research questions relating to individual factors and an open-source code base with which to analyze these data. Many of the transcription elongation regulators studied here are conserved in all domains of life, as are many of the transcriptional phenotypes we examined, including antisense transcription and Pol II pausing. These insights into transcription regulation in *S. cerevisiae* will serve as a foundation for learning more about transcription in multicellular eukaryotes.

# Materials and methods
## Yeast mutant generation
To create deletion mutants of the 41 factors analyzed, the parent strain YSC001 (BY4741 *rpb3::rpb3-3xFLAG NAT*) (*Churchman and Weissman, 2011*) was transformed with PCR products of the HIS3 gene flanked by 40 bp of homology upstream and downstream of the start and stop codons for the gene of interest. Standard lithium acetate transformations were used.

## NET-seq library generation

Cultures for NET-seq were prepared as described in *Churchman and Weissman, 2012*. Briefly, overnight cultures from single yeast colonies grown in Yeast Extract–Peptone–Dextrose (YPD) medium were diluted to $OD_{600}$=0.05 in 1 L of YPD medium and grown at 30°C shaking at 200 rpm until reaching an $OD_{600}$=0.6–0.8. Cultures were then filtered over 0.45-mm pore size nitrocellulose filters (Whatman). Yeast was scraped off the filter with a spatula pre-chilled in liquid nitrogen and plunged directly into liquid nitrogen as described in *Churchman and Weissman, 2012*. Mixer mill pulverization was performed using the conditions described above for six cycles. NET-seq growth conditions, immunoprecipitations, and isolation of nascent RNA and library construction were carried out as described in *Churchman and Weissman, 2012*. A random hexamer sequence was added to the linker to improve ligation efficiency and allow for the removal of any library biases generated from the RT step as described in *Mayer et al., 2015*. After library construction, the size distribution of the library was determined by using a 2100 Bioanalyzer (Agilent), and library concentrations were determined by Qubit 2.0 fluorometer (Invitrogen). 3' end sequencing of all samples was carried out on an Illumina NextSeq 500 with a read length of 75 bp. For analysis of *cac1Δ*, *cac2Δ*, and *cac3Δ*, raw Fastq files were obtained from *Marquardt et al., 2014* and re-aligned using the parameters described below.

## Processing and alignment of NET-seq data

The adapter sequence (ATCTCGTATGCCGTCTTCTGCTTG) was removed using cutadapt with the following parameters: -O 3 m 1 `--length-tag` 'length=.' Raw fastq files were filtered using PRINSEQ (http://prinseq.sourceforge.net/) with the following parameters: -no_qual_header -min_len 7 min_qual_mean 20 -trim_right 1 -trim_ns_right 1 -trim_qual_right 20 -trim_qual_type mean -trim_qual_window 5 -trim_qual_step 1. Random hexamer linker sequences (the first six nucleotides at the 5' end of the read) were removed using custom Python scripts, but remained associated with the read. Reads were then aligned to the SacCer3 genome obtained from the *Saccharomyces* Genome Database using the TopHat2 aligner (*Kim et al., 2013*) with the following parameters: `--read-mismatches` 3 `--read-gap-length` 2 `--read-edit-dist` 3 `--min-anchor-length` 8 `--splice-mismatches` 1 `--min-intron-length` 50 `--max-intron-length` 1200 `--max-insertion-length` 3 `--max-deletion-length` 3 `--num-threads` `--max-multihits` 100 `--library-type` fr-firststrand `--segment-mismatches` 3 `--no-coverage-search` `--segment-length` 20 `--min-coverage-intron` 50 `--max-coverage-intron` 100000 `--min-segment-intron` 50 `--max-segment-intron` 500000 `--b2-sensitive`. To avoid any biases toward favoring annotated regions, the alignment was performed without providing a transcriptome. RT mispriming events were identified and removed where molecular barcode sequences correspond exactly to the genomic sequence adjacent to the aligned read. With NET-seq, the 5' end of the sequencing, which corresponds to the 3' end of the nascent RNA fragment, is recorded with a custom Python script using the HTSeq package (*Anders et al., 2015*). NET-seq data were normalized by million mapped reads. Replicate correlations were performed comparing RPKM of each gene in each replicate; replicates were considered highly correlated with a Pearson correlation of $R^2$ ≥0.75. Raw NET-seq data of highly correlated replicates were merged, and then re-normalized by million mapped reads. For analysis of *rco1Δ*, raw Fastq files were obtained from *Churchman and Weissman, 2011* and re-aligned using the parameters described, except without removal of hexamer sequences.

## Differential gene transcription and gene ontology enrichment analysis

Differential transcription analysis between deletion strains (two replicates each) and wild-type strains (four replicates) was performed using DESeq2 (*Love et al., 2014*) for all sense transcription units annotated in *Xu et al., 2009*. To account for antisense transcription, matching antisense transcription units were added to the annotation, as long as they did not overlap with a known sense gene. These added antisense transcription units were ignored in reporting the number of differentially expressed genes (*Figure 1B and C*; *Figure 1—figure supplement 1B*). Genes were considered differentially transcribed if they had an adjusted p-value <0.05 and an absolute $log_2$-fold change >1.0.

GO term enrichment analysis was performed with The Ontologizer (http://ontologizer.de/intro/) (*Bauer et al., 2008*; *Grossmann et al., 2007*; *Ashburner et al., 2000*; *The Gene Ontology Consortium, 2019*) using the parent-child analysis method. The GO term 'antisense transcription' (GO: 9999999) was added to the go.obo file, and this new GO term was associated with all antisense

transcription units described above by modifying the file sdg.gaf. Fold enrichment and adjusted p-value for each GO by deletion strain pair are reported in *Supplementary file 3*.

## Antisense transcription

For analysis of antisense transcription, the coordinates of protein-coding transcription units from *Xu et al., 2009* were reversed and annotated as 'antiXXXX', where 'XXXX' is the name of the gene encoded on the sense strand. Those that overlapped known sense transcription units were removed. This expanded annotation file was used to produce read count tables for DESeq2. To generate antisense heatmaps, the $\log_2$ RPKM of NET-seq reads was used. Analysis at coding genes ranged from 250 bp upstream of the TSS to 4000 bp downstream of the coding TSS. To allow comparison between mutant and wild-type samples, a pseudocount of 1 was added to every position in all samples before calculating the $\log_2$ RPKM. Differential heatmaps were calculated by taking the $\log_2$ ratio of mutant/wild-type RPKM at each position.

## Pausing index calculation

Pausing indices were calculated as the length-normalized Pol II density in the region of interest (–50 bp to +150 bp around TSS, ±100 bp around poly(A) sites, and ±10 bp around 5' and 3' splice sites) divided by the length-normalized Pol II density in the remainder of the gene, as illustrated in *Figure 3G*.

## Metagene analysis

Only protein-coding, non-overlapping genes were included in the metagene analysis. The regions analyzed were –100 to +600 bp surrounding the most abundant TSS, –500 to +200 bp surrounding poly(A) sites, as identified in *Pelechano et al., 2013*, and ±25 bp surrounding annotated 3' and 5' splice sites. NET-seq signal across each region was normalized, and the Loess smoothed mean (span = 0.01) and 95% confidence interval are plotted for NET-seq generated from each deletion strain across each region of interest.

## Splicing index calculation

Cac2Δ and wild-type RNA-seq data were retrieved from *Hewawasam et al., 2018* under the GEO accession number GSE98397. Splicing index calculations were determined for each gene by counting the number of reads that span exon junctions by at least three nucleotides and measuring the number of spliced reads divided by unspliced reads; splicing index = 2 * spliced reads/(5' SS unspliced + 3' SS unspliced reads) as in *Drexler et al., 2020*.

## Extracting pause positions

Pauses were identified in previously annotated transcription units (*Xu et al., 2009*) of well-expressed genes (average of >2 reads per base-pair in two replicates). Pauses were defined as having reads higher than three standard deviations above the mean of the surrounding 200 nucleotides which do not contain pauses. Mean and standard deviation were calculated from a negative binomial distribution fit to the region of interest. Pauses were required to have at least two reads regardless of the gene's sequencing coverage. Our analysis algorithm for identifying pause sites uses the IDR analysis, which is the standard for analyzing ENCODE ChIP-seq data (*Li et al., 2011*; *Landt et al., 2012*). Here, many pause sites are identified in each replicate and ranked. The peaks in each biological replicate are compared, starting with the strongest. When the ranks of the peaks stop corresponding, a transition point is identified and the lower ranked peaks are marked as irreproducible. The methodology does not require an arbitrary cutoff, and all pause sites are considered in the comparison between replicates, reducing false negatives. Pauses were considered reproducible and used in downstream analyses when the IDR is <1% between two replicates. To calculate the IDR of each pause, $\log_{10}$ of pause strength (number of reads in pause) for each replicate was used as a proxy for pause score. IDR was calculated using the est.IDR function of the idr R package (mu = 3, sigma = 1, rho = 0.9, p=0.5) (*Li et al., 2011*). Reproducible pauses were visualized using the IGV genome browser (*Robinson et al., 2011*). Because the *cac1Δ*, *cac2Δ*, and *cac3Δ* strains were constructed by a different lab (*Marquardt et al., 2014*), these strains were excluded from these analyses. Additionally *gcn5Δ* was excluded because of low sequencing coverage resulting in only 15 genes passing the coverage threshold.

## Pol II pausing location and strength

Pause density was calculated as the ratio of total number of pauses to the total length of the genome considered when extracting pause positions (combined length of all well-expressed genes in both replicates of each deletion strain). To identify deletions that induced similar pausing patterns, 8644 pauses were found to be shared in at least eight strains and in regions sufficiently covered in multiple deletion strains. Shared pauses were visualized with a heatmap, clustered on both axes using the eisenCluster correlation clustering method in the hybridHclust R package (*Chipman and Tibshirani, 2006*), which takes into account missing data (where there was not enough coverage to confidently identify pausing in a particular deletion strain). Similarity in pause loci was also visualized as a scatter plot of the first two principal components. When calculating distribution of pauses across the gene body, all genes in which pauses were identified were normalized in length; the 5′ gene region was defined as the first 15% of each gene, the mid-gene region was defined as extending from the 15th percentile of gene length to the 85th percentile, and the 3′ gene region was defined as starting at 85% of gene length and extending to the annotated poly(A) site. The scrambled control for the pausing location analysis was created by randomly scrambling all identified pauses in all deletion strains across the gene in which they were discovered.

## Pol II pause loci sequence motifs

All analyses related to sequence motifs underlying pause loci were conducted using the MEME suite of tools (*Bailey et al., 2009*; *Bailey and Elkan, 1994*). The sequence ±10 bp around each identified, reproducible pause (as well as the matched scrambled control) was extracted and used to run the MEME tool using parameters to find 0–1 motif per sequence, motifs 6–21 bp in length, and up to 10 motifs with an *E*-value significance threshold of 0.05 (*Bailey and Elkan, 1994*). These significant motifs were compared to known transcription factor binding site motifs in the YEASTRACT_20130918 database (*Teixeira et al., 2014*) using the TOMTOM tool (*Gupta et al., 2007*) using default parameters, calling all hits as significant with an *E*-value greater than 0.1. TOMTOM searches were only performed on those motifs with a relative entropy greater than five and only the top match is reported.

## Random forest classifier for Pol II pausing loci

The predictive value of chromatin and DNA features for identifying Pol II pause loci was determined using a random forest model with the randomForest R package (*Breiman, 2001*). All reproducible Pol II pause loci were included in these analyses, as were an equal number of shuffled control loci. The shuffled control loci were selected to maintain the same number of real and control loci in each gene, controlling for effects of differential gene expression. In total, 51 chromatin and DNA features were compiled for all pause loci (*Supplementary file 5*; *Chiu et al., 2016*; *Oberbeckmann et al., 2019*; *Pelechano et al., 2013*; *Turner and Mathews, 2010*; *Umeyama and Ito, 2018*; *Vinayachandran et al., 2018*; *Weiner et al., 2015*). Before applying the random forest classifier, we examined the distribution of values for each numeric feature (not discrete sequence) for real Pol II pauses compared to the scrambled control loci; statistical significance in the difference between these distributions was calculated with a Student's t-test, correcting for multiple hypothesis testing with the Bonferroni correction. From the random forest classifier, feature importance scores were generated using a random forest classifier with 75% training and 25% testing sets; for wild-type yeast, this is 10,495 training and 3499 training loci. Due to the low number of reproducible pauses identified in the *gcn5Δ* deletion strain, it was excluded from these analyses.

Reported feature importance values are the mean decreases of accuracy over all out-of-bag cross-validated predictions, when a given feature is permuted after training, but before prediction. Optimized parameters were selected for random forest classifiers trained using all features (*Figure 5—figure supplement 1B*):ncat = 4, mtry = 20, ntrees = 2500. ROC curve and AUC measurements were determined from binary prediction probabilities and calculated using the ROCR R package (*Sing et al., 2005*). Prediction accuracy was determined by measuring the difference between the model's predictions on a held-out test set and measured variables. The baseline score was determined using a 'null' parameter that has the same value for every training and testing pair; thus, baseline represents the prediction accuracy with no additional information added to the model. To assess the transferability of a random forest classifier trained on Pol II pause loci in one strain, a model was trained on

100% of real and shuffled control Pol II loci from one deletion strain and then tested on all those pause loci in a second deletion strain, which was not included in the training set.

## Code availability

All scripts and data analyses are available at https://github.com/churchmanlab/Yeast_NETseq_Screen; *Couvillion and Churchman Lab, 2022*. All plots were created in R using ggplot2 (*R Development Core Team, 2013*; *Wickham, 2016*).

## Acknowledgements

We thank S Issac and C Patil for constructive feedback on the manuscript. This work was supported by National Institutes of Health grant R01-HG007173 (LSC) and a Ruth L Kirschstein National Research Service Award F31 HG010570 (KCL).

## Additional information

### Competing interests

L Stirling Churchman: Reviewing editor, *eLife*. The other authors declare that no competing interests exist.

### Funding

| Funder | Grant reference number | Author |
| --- | --- | --- |
| National Institutes of Health | R01-HG007173 | L Stirling Churchman |
| National Institutes of Health | F31 HG010570 | Kate C Lachance |

The funders had no role in study design, data collection and interpretation, or the decision to submit the work for publication.

### Author contributions

Mary Couvillion, Formal analysis, Software, Validation, Visualization, Writing - review and editing; Kevin M Harlen, Data curation, Formal analysis, Investigation, Methodology, Supervision; Kate C Lachance, Formal analysis, Software, Visualization, Writing - original draft; Kristine L Trotta, Erin Smith, Christian Brion, Brendan M Smalec, Data curation; L Stirling Churchman, Conceptualization, Supervision, Writing - review and editing

### Author ORCIDs

Kristine L Trotta http://orcid.org/0000-0002-8166-7696
L Stirling Churchman http://orcid.org/0000-0003-3888-2574

### Decision letter and Author response

Decision letter https://doi.org/10.7554/eLife.78944.sa1
Author response https://doi.org/10.7554/eLife.78944.sa2

## Additional files

### Supplementary files

• Supplementary file 1. Pairwise correlation between all replicates included in reverse genetic screen.

• Supplementary file 2. Differential transcription of each gene across all deletion strains. Lists every gene differentially transcribed, both sense and antisense strands, as determined using DESeq2 (*Love et al., 2014*), for every deletion strain included in screen. Significance was determined to be those genes with an adjusted -value ≤ 0.05 and an absolute log2(fold change) in expression compared to wild-type ≥1. For each significantly differentially transcribed gene, the log2(fold

change) and adjusted p-value is reported.

• Supplementary file 3. Differentially transcribed genes are enriched for GO terms. This table lists all GO terms that were significantly enriched in at least one deletion strain. For each GO term, if it was found to be significant in a given deletion strain, the fold enrichment and adjusted p-value (in parentheses) are listed. This table is separated into three sheets: those GO terms derived from either significantly up- or downregulated genes (purple), only significantly downregulated genes (red), and only significantly upregulated genes (blue).

• Supplementary file 4. Significant motifs underlying pauses across deletion strains with transcription factor binding site matches.

• Supplementary file 5. Sources of chromatin features used in random forest classifier.

• Supplementary file 6. Results of t-test between distributions of feature values comparing real and shuffled control pauses. For each numeric chromatin feature, the t-value, p-value, and indication of significance is given resulting from a Student's t-test comparing the distribution of values surrounding real and shuffled control pauses. Table corresponds to boxplots illustrating distributions for all numeric chromatin features (*Figure 5—figure supplement 1*). Significance indicators are applied after a Bonferroni correction for multiple hypotheses (*<0.05, **<0.01, ***<0.001).

• MDAR checklist

## Data availability

The accession number for the Illumina sequencing reported in this paper is Gene Expression Omnibus (GEO): GSE159603.

The following dataset was generated:

| Author(s) | Year | Dataset title | Dataset URL | Database and Identifier |
|---|---|---|---|---|
| Couvillion MT, Lachance KC, Harlen KM, Trotta KL, Smith E, Churchman LS | 2021 | Dynamics of transcription elongation are finely-tuned by dozens of regulatory factors | https://www.ncbi.nlm.nih.gov/geo/query/acc.cgi?acc=GSE159603 | NCBI Gene Expression Omnibus, GSE159603 |

The following previously published dataset was used:

| Author(s) | Year | Dataset title | Dataset URL | Database and Identifier |
|---|---|---|---|---|
| Hewawasam GS, Dhatchinamoorthy K, Mattingly M, Seidel C | 2017 | Chromatin assembly factor-1 (CAF-1) chaperone regulates Cse4 deposition at active promoter regions in budding yeast | https://www.ncbi.nlm.nih.gov/geo/query/acc.cgi?acc=GSE98397 | NCBI Gene Expression Omnibus, GSE98397 |

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
