## [Editor Report]

In this manuscript the authors have conducted native elongation transcript sequencing on yeast strains deleted for one of 41 different transcription, chromatin modifying and RNA processing factors. They find that a large fraction of these deletions affect transcription elongation and RNA Pol II pausing indicating that elongation is carefully regulated by many factors.

---

## [Decision Letter]

**Decision letter after peer review:**

[Editors’ note: the authors submitted for reconsideration following the decision after peer review. What follows is the decision letter after the first round of review.]

Thank you for submitting the paper "Dynamics of transcription elongation are finely tuned by dozens of regulatory factors" for consideration by *eLife*. Your article has been reviewed by 3 peer reviewers, including Jerry L Workman as the Reviewing Editor and Reviewer #1, and the evaluation has been overseen by a Senior Editor.

Comments to the Authors:

We are sorry to say that, after consultation with the reviewers, we have decided to reject this manuscript. There are technical issues noted in the review below that would need to be resolved to support many of the conclusions. If these issues can be addressed in a satisfactory way we would welcome submission of a new version of the manuscript.

In this study, "Dynamics of transcription elongation are finely tuned by dozens of regulatory factors" the authors present an impressive amount of native elongating transcript sequencing (NET-Seq) data and perform in-depth analysis of the dataset. Overall, the focus of this work was to determine the contributions of 41 transcription/chromatin related non-essential gene products to RNA Polymerase II transcription at different phases of transcription. This includes in-depth characterization of RNA Polymerase II pausing in each deletion strain and an analysis of sense and antisense transcription events. The introduction, which sets up the goals of the study, was very descriptive of transcription in general and lacked some focus discussing events that occur in multiple biological systems although this study was performed using yeast as the sole model system. It is stated that it is currently unknown how Pol II pausing contributes to gene expression levels however it could also be argued that Pol II pausing is, by nature, inhibitory to transcript production. Antisense transcription in yeast has also been shown by others to be inhibitory to sense transcription in multiple contexts including different yeast deletion backgrounds.

The Churchman lab are leading experts in NET-Seq method development and data analysis and it is likely that the data produced for this study are of high quality. The major weakness in the context of this current study is that this study is NET-Seq focused with a lack of follow up experiments. This concern is partially mitigated by the breadth of the work that was performed. However, some data focuses on the reproducibility of specific events, such as Pol II pausing, and only two replicates were performed for each mutant. In fact, Figure S5A suggests that pausing reproducibility across the two replicates may be poor. Figures 4-7 focus on this pause data so lack of reproducibility of this measurement is a major concern.

The data presented covers many of the 41 mutant strains that were used in the study. It does a nice job of describing both extremes of changes for each aspect that is discussed relative to the parental yeast strain. The study often references data from other studies to suggest potential interpretation of the results but no major follow up studies were performed to provide strength to these interpretations or glean new mechanism. Many of the findings support prior studies by others making this a useful resource yet not necessarily providing many novel insights. The uncertainly regarding the pause site reproducibility limits the potential impact of that portion of the work.

Recommendations for the author:

The manuscript by Couvillion, Harlen, Lachance et al., describes the effect of deleting a set of elongation-related factors on Pol II pausing and Pol II antisense transcription using NET-seq in budding yeast. Pausing and antisense transcription were extensively compared between genes/regions and between strains. Overall, the work generated mainly expected results but did not highlight any clearly new concept or finding. One unexpected observation is that deletion of subunits of the CAF-I histone chaperone led to increased pausing near splice sites but this observation was not pursued further.

Although no major breakthrough came out of this work, the dataset generated in this study represents a valuable resource for the "transcription community" (notwithstanding a concern described below).

Besides the lack of a major breakthrough, enthusiasm for the work in its current form is dampened mainly by the first two issues detailed below:

1) I am concerned that the conclusions about pausing might be mitigated by noise in the pause site calls. First, I was surprised to see that in most cases, deleting elongation factor genes led to decreased pausing. Intuitively, I would have expected elongation factors to help suppress pauses, not promote them. This is notably unexpected for the spt4 mutant since Spt4 has been clearly shown to suppress pausing, yet the NET-seq data suggest the opposite.

This peculiar observation (which is not commented on by authors) raised some suspicions about the pause site calls. Scrutinizing the NET-seq literature quickly revealed that NET-seq peaks can often occur consequent to technical artifacts (RNA processing intermediates, PCR duplicates, products of mispriming during RT, etc.). The Mayer lab recently published a version of NET-seq that limits these artifacts (https://doi.org/10.1093/nar/gkab208). Using this protocol, the Mayer lab found that mammalian Pol II pauses every 3,000-30,000 nucleotides. This is far less frequent than the 31 nucleotides suggested in the current work. While this may reflect differences between species, this reinforced my suspicions about these pause site calls. The sequence bias around paused sites is also different in the current study compared to previous work in mammals and *E. coli*, further suggesting that the current study might include a large number of artifactual pause site calls.

Can the authors comment on the possibility that some (perhaps a lot) of their called pause sites are not bona fide, and to what extent this might have affected their conclusions? Is it possible for them to leverage some of the improvements described in the Mayer paper to test whether this would affect some of their key conclusions?

2) I am concerned about the use of the antisense/sense ratio as a measure of antisense transcription. This is a convoluted measure that is affected both by changes in sense and antisense transcription. Hence, a change in the antisense/sense ratio simply can not be assimilated to an effect on antisense transcription; it may just as well reflect effects on sense transcription or a combination of both sense and antisense.

This mitigates several of the conclusions made by the authors. For example, on p.11: "This result implies that strong antisense pausing suppresses antisense transcription, perhaps by promoting termination and thereby preventing antisense transcription deep into gene bodies". This conclusion is mitigated by the use of antisense/sense ratio as a measure of antisense transcription. It appears just as possible that strong antisense pausing stimulates sense transcription.

Similarly, on p.18: "Indeed, differentially transcribed genes showed pronounced changes in their antisense:sense transcription ratios, especially for a subset of sensitive genes that are differentially transcribed in many of the deletion strains". By definition "differentially transcribed genes" means that sense transcription is affected. This alone will affect the antisense:sense ratio.

A measure of the absolute antisense transcription levels in WT and mutant strains should be attempted. While it may be difficult to compare such measurement across strains, it would - in principle - be a more accurate measure of antisense transcription. I suspect that most conclusions will remain, but the current analysis is simply not sound.

3) Figures 1 and 2 are quite descriptive and have some presentation challenges. For instance Figure 2D, E, & F appear to show very subtle changes. In the scale used for those figures it is difficult to see the changes that are occurring. It is recommended that a smaller range be used so that the changes can be more clearly visualized. Many of the changes have been previously reported although not using NET-Seq analysis to my knowledge. In these cases the NET-Seq data could be used as a higher quality resource and perhaps that aspect could be discussed (advantages, etc.).

4) Figure 3 presents some interesting data that are novel to my knowledge. These novel findings, such as the contribution of CAF-I to Pol II density changes at splice sites, should be discussed in more depth to increase the novelty of the work.

Other comments:

a) The title seems inappropriate. "Dynamics of transcription elongation" suggest that elongation parameters (speed and processivity) were assayed, which is not the case. Instead, the paper focuses on pausing and anti-sense transcription. While these phenomena are linked to elongation, this does not justify the current title.

b) I am surprised that histone chaperones notoriously linked to elongation (e.g. Spt6, FACT, Spt2, etc.) were not included in this study.

c) The abstract mentions co-transcriptional processing (presumably RNA processing). Yet, I do not think that RNA processing was monitored in this study (except perhaps for analyzing a published dataset for CAF-I).

d) check spelling of all forms of the word (and processes related to) ubiquitin. There are multiple spellings/typos.

e) Features for the AI modeling are described as "chromatin features" but use features both within and outside of chromatin considerations. I would consider renaming.

f) There is a missed opportunity for more in-depth discussion of transcription factor contribution to potential pause sites and for discussion of potential RNA binding protein contributions.

[Editors’ note: further revisions were suggested prior to acceptance, as described below.]

Thank you for resubmitting your work entitled "Transcription elongation is finely tuned by dozens of regulatory factors" for further consideration by *eLife*. Your revised article has been evaluated by James Manley (Senior Editor) and a Reviewing Editor.

The reviewers have discussed their reviews with one another, and the Reviewing Editor has drafted this to help you prepare a revised submission. As this is a complicated paper with varying effects of different deletions etc. the reviewers thought that some issues need to be clarified in the text to make absolutely clear what robust conclusions can be drawn from this study.

Essential revisions:

1) The revised manuscript by Couvillion, Harlen, Lachance et al. is vastly improved. The authors have adequately addressed my main concerns. The only aspect that remains unclear to me concerns the fact that mutants for elongation factors such as dst1D and spt4D lead to decreased number of paused sites (Figure 4). As stated by the authors, this is unexpected since these factors are known to prevent (or help alleviate) pausing. Consistent with this expected behavior (and in apparent contradiction with the analysis shown in Figure 4), dst1D cells harbor a clear increase in Pol II density in the 5' region. This was highlighted in my initial review and the authors have addressed this by adding some speculations on page 12. This explanation, however, is vague and not compelling. One possible explanation would be that, in strains such as dst1D, pause sites are fewer but stronger. In this scenario, Pol II would pause less often but have a harder time getting out of the pause state in dst1D (and others) cells. Does the data allow testing this possibility? I feel that the manuscript would benefit from straightening that aspect.

2) For the most part my major concerns have been addressed. The reproducibility of the experiments was carefully assessed (Figure S5A & B) and the use of the irreproducibility discovery rate (IDR) with clear cutoffs sets clear quantitative standards for each dataset. It is clear that some of the knockout strains have a low overall impact on pausing and this is discussed through comparison of the median TSS pausing index.

3) Much of my major concern is with the use of a discussion of machine learning being used for prediction of pausing location. The machine learning section appears to more clearly provide new models for the contribution of different DNA/chromatin features to changes in pausing observed in any individual elongation factor deletion strain. This point can be addressed through writing to clarify what the machine learning analysis actually provides rather than what it could potentially do (predict pause sites) but appeared to fall short of.

4) I appreciate the care taken to address the concerns raised about the sense/antisense ratio analysis from the initial manuscript. This clarity of this section is much improved. I have a comment regarding this specific statement:

"The factors whose deletions led to the largest increase in the antisense transcription were those involved in the regulation of histone acetylation, including members of the Rpd3S-Set2 pathway (Set2) and the major histone H4 acetyltransferase complex NuA4 (Eaf1), emphasizing the role of acetylation in antisense transcription (Carrozza et al., 2005; Churchman and Weissman, 2011; Krogan et al., 2003; Murray et al., 2015; Murray and Mellor, 2016)."

For this statement, the deletion of an acetyltransferase will decrease acetylation whereas the deletion of members of the Rpd3S-Set2 pathway increase acetylation. As a consequence it is recommended to state "emphasizing the role of acetylation / deacetylation in antisense transcription" for clarity.

---

## [Author Response]

[Editors’ note: the authors resubmitted a revised version of the paper for consideration. What follows is the authors’ response to the first round of review.]

In this study, "Dynamics of transcription elongation are finely tuned by dozens of regulatory factors" the authors present an impressive amount of native elongating transcript sequencing (NET-Seq) data and perform in-depth analysis of the dataset. Overall, the focus of this work was to determine the contributions of 41 transcription/chromatin related non-essential gene products to RNA Polymerase II transcription at different phases of transcription. This includes in-depth characterization of RNA Polymerase II pausing in each deletion strain and an analysis of sense and antisense transcription events. The introduction, which sets up the goals of the study, was very descriptive of transcription in general and lacked some focus discussing events that occur in multiple biological systems although this study was performed using yeast as the sole model system. It is stated that it is currently unknown how Pol II pausing contributes to gene expression levels however it could also be argued that Pol II pausing is, by nature, inhibitory to transcript production. Antisense transcription in yeast has also been shown by others to be inhibitory to sense transcription in multiple contexts including different yeast deletion backgrounds.

The introduction has been edited to focus on yeast-specific results and the regulation of transcription elongation. We also clarified our discussion of Pol II pausing and antisense transcription (pg. 3-5).

The Churchman lab are leading experts in NET-Seq method development and data analysis and it is likely that the data produced for this study are of high quality. The major weakness in the context of this current study is that this study is NET-Seq focused with a lack of follow up experiments. This concern is partially mitigated by the breadth of the work that was performed. However, some data focuses on the reproducibility of specific events, such as Pol II pausing, and only two replicates were performed for each mutant. In fact, Figure S5A suggests that pausing reproducibility across the two replicates may be poor. Figures 4-7 focus on this pause data so lack of reproducibility of this measurement is a major concern.

Pol II pause sites that are not reproducible in NET-seq data arise due to stochastic technical fluctuations and biological fluctuations, which are especially pronounced at the single nucleotide level. To remove the pause sites that do not correspond across replicates, we used a stringent irreproducible discovery rate (IDR) analysis (IDR of 1%). The approach increases the number of pause sites considered initially, does not require an arbitrary threshold for pause site calling, and reduces false negatives. Thus, the irreproducible pause sites in Figure S5A do not represent a failure of the methodology, rather they arise as a result of the IDR analysis. Nevertheless, we understand the reviewer’s concern about the reproducibility of our pause sites, and we have added an additional analysis that compares our results across pairs of replicates, demonstrating that the majority of reproducible pauses overlap across sets of replicates (Figure S5B). We edited the text in the Results (pg. 11) and Methods (pgs. 21-22) to explain our approach more clearly.

The data presented covers many of the 41 mutant strains that were used in the study. It does a nice job of describing both extremes of changes for each aspect that is discussed relative to the parental yeast strain. The study often references data from other studies to suggest potential interpretation of the results but no major follow up studies were performed to provide strength to these interpretations or glean new mechanism. Many of the findings support prior studies by others making this a useful resource yet not necessarily providing many novel insights. The uncertainly regarding the pause site reproducibility limits the potential impact of that portion of the work.Recommendations for the author:The manuscript by Couvillion, Harlen, Lachance et al., describes the effect of deleting a set of elongation-related factors on Pol II pausing and Pol II antisense transcription using NET-seq in budding yeast. Pausing and antisense transcription were extensively compared between genes/regions and between strains. Overall, the work generated mainly expected results but did not highlight any clearly new concept or finding. One unexpected observation is that deletion of subunits of the CAF-I histone chaperone led to increased pausing near splice sites but this observation was not pursued further.Although no major breakthrough came out of this work, the dataset generated in this study represents a valuable resource for the "transcription community" (notwithstanding a concern described below).Besides the lack of a major breakthrough, enthusiasm for the work in its current form is dampened mainly by the first two issues detailed below:1) I am concerned that the conclusions about pausing might be mitigated by noise in the pause site calls. First, I was surprised to see that in most cases, deleting elongation factor genes led to decreased pausing. Intuitively, I would have expected elongation factors to help suppress pauses, not promote them. This is notably unexpected for the spt4 mutant since Spt4 has been clearly shown to suppress pausing, yet the NET-seq data suggest the opposite.

We thank the reviewer for highlighting this possible point of confusion. In this study, we performed two types of analyses to look at Pol II pausing. The first is quantifying peaks of Pol II at different positions in the gene body (transcription start sites, splice sites etc.). The other is the precise location on the DNA where Pol II prefers to pause. We found a strong trend for elongation factors to impact the peaks of Pol II at the start of genes near transcription start sites, which is consistent with their established roles. Where Pol II prefers to pause on DNA (i.e pause sites) is a different question and won’t necessarily depend on elongation factors. Pause site densities include only the locations at which Pol II typically pauses in many cells, so it is not a measure of the absolute frequency of Pol II pausing. In addition, the densities are not related to the Pol II catalysis rate. So, these transcription elongation factors may facilitate other aspects of transcription elongation or they only act locally to influence Pol II during specific points of regulation. We now include this discussion in the text (pg. 10,12).

This peculiar observation (which is not commented on by authors) raised some suspicions about the pause site calls. Scrutinizing the NET-seq literature quickly revealed that NET-seq peaks can often occur consequent to technical artifacts (RNA processing intermediates, PCR duplicates, products of mispriming during RT, etc.). The Mayer lab recently published a version of NET-seq that limits these artifacts (https://doi.org/10.1093/nar/gkab208). Using this protocol, the Mayer lab found that mammalian Pol II pauses every 3,000-30,000 nucleotides. This is far less frequent than the 31 nucleotides suggested in the current work. While this may reflect differences between species, this reinforced my suspicions about these pause site calls. The sequence bias around paused sites is also different in the current study compared to previous work in mammals and *E. coli*, further suggesting that the current study might include a large number of artifactual pause site calls.

We performed our pause site analysis on genes that were highly covered by NET-seq reads, which is possible in yeast where the nascent transcriptome is relatively small.

Consistently, our pause site densities were not sensitive to the sequencing depth of our libraries (Figure S5F). By contrast, it is too costly to sequence human NET-seq libraries deeply enough to determine all pause sites for many genes. Rather, the Mayer group searched for pauses in all genes regardless of coverage level, and the number of pauses identified was sensitive to the number of aligned reads used in the analysis. Thus, as they note, the number of pause sites they report is underestimated. Interestingly, they describe some genes where Pol II pause sites were identified quite frequently (every 146 nt), presumably due to high NET-seq coverage at those loci. The sequence motif differences between yeast and human pause sites could be due to species differences or the different subsets of genes analyzed in the two studies.

Can the authors comment on the possibility that some (perhaps a lot) of their called pause sites are not bona fide, and to what extent this might have affected their conclusions? Is it possible for them to leverage some of the improvements described in the Mayer paper to test whether this would affect some of their key conclusions?

We thank the reviewers for raising this critical point. Reverse transcription mispriming occurs when the RT primer associates internally within the nascent RNA instead of the oligo ligated to the 3’ end. Due to the large human genome and the long pre-mRNAs (~18,000 nt on average), these events occur more frequently in human NET-seq data. They are easily identifiable computationally because reads from mispriming events do not contain a unique molecular identifier that is added during the oligo ligation step. Furthermore, mispriming reads align proximal to a sequence that is the reverse complement of the start of the RT primer. We already remove these reads from our NET-seq analysis, although very few arise in our yeast NET-seq data. We were pleased when the Mayer lab developed the nested NET-seq protocol, because it is useful for our projects generating human NET-seq libraries. We did not expect the nested approach to impact our yeast data due to the relatively small yeast transcriptome. Nevertheless, considering the emphasis on pause sites in this study, we agree with the reviewers that we should determine whether the nested approach impacts yeast NET-seq data and pause site determination. We have now compared wild-type yeast data from libraries prepared using nested NET-seq to those prepared with the standard protocol. We found that in yeast, the genome is so small that using the nested NET-seq library approach does not change the number of pauses identified (Figure S5C) nor does it decrease the fraction of pause sites with adapter-like sequence downstream, which is expected at sites of mispriming (Figure S5D). Results of this analysis are now described on pages 11-12.

2) I am concerned about the use of the antisense/sense ratio as a measure of antisense transcription. This is a convoluted measure that is affected both by changes in sense and antisense transcription. Hence, a change in the antisense/sense ratio simply can not be assimilated to an effect on antisense transcription; it may just as well reflect effects on sense transcription or a combination of both sense and antisense.This mitigates several of the conclusions made by the authors. For example, on p.11: "This result implies that strong antisense pausing suppresses antisense transcription, perhaps by promoting termination and thereby preventing antisense transcription deep into gene bodies". This conclusion is mitigated by the use of antisense/sense ratio as a measure of antisense transcription. It appears just as possible that strong antisense pausing stimulates sense transcription.Similarly, on p.18: "Indeed, differentially transcribed genes showed pronounced changes in their antisense:sense transcription ratios, especially for a subset of sensitive genes that are differentially transcribed in many of the deletion strains". By definition "differentially transcribed genes" means that sense transcription is affected. This alone will affect the antisense:sense ratio.A measure of the absolute antisense transcription levels in WT and mutant strains should be attempted. While it may be difficult to compare such measurement across strains, it would - in principle - be a more accurate measure of antisense transcription. I suspect that most conclusions will remain, but the current analysis is simply not sound.

We thank the reviewer for the suggestion. We now analyzed antisense transcription without using a ratio to sense transcription. We achieved this directly using the statistical gene expression analysis package DEseq2 by annotating all the antisense transcripts. Furthermore, in the reanalysis of our data with DEseq2, we found fewer sensitive genes that were impacted by the removal of regulatory factors, and we decided to remove this analysis from the manuscript. These new analyses led to reformatted Figures 2 and S2 that combines the previous Figures 2 and 7.

3) Figures 1 and 2 are quite descriptive and have some presentation challenges. For instance Figure 2D, E, & F appear to show very subtle changes. In the scale used for those figures it is difficult to see the changes that are occurring. It is recommended that a smaller range be used so that the changes can be more clearly visualized.

We apologize, but we are unclear about what the reviewer is referring to. On our computer screens and printouts, the changes appear clearly.

Many of the changes have been previously reported although not using NET-Seq analysis to my knowledge. In these cases the NET-Seq data could be used as a higher quality resource and perhaps that aspect could be discussed (advantages, etc.).

We have edited the text to clarify where NET-seq data provides higher quality views of previously reported trends (pg. 8).

4) Figure 3 presents some interesting data that are novel to my knowledge. These novel findings, such as the contribution of CAF-I to Pol II density changes at splice sites, should be discussed in more depth to increase the novelty of the work.

We now discuss these findings in greater depth in the Discussion (pg. 17-18).

Other comments:a) The title seems inappropriate. "Dynamics of transcription elongation" suggest that elongation parameters (speed and processivity) were assayed, which is not the case. Instead, the paper focuses on pausing and anti-sense transcription. While these phenomena are linked to elongation, this does not justify the current title.

We have changed the title to “Transcription elongation is finely tuned by dozens of regulatory factors.”

b) I am surprised that histone chaperones notoriously linked to elongation (e.g. Spt6, FACT, Spt2, etc.) were not included in this study.

In this study, we only included non-essential factors. Unfortunately, many histone chaperones, including the ones mentioned, are essential, or their absence leads to extremely slow growth.

c) The abstract mentions co-transcriptional processing (presumably RNA processing). Yet, I do not think that RNA processing was monitored in this study (except perhaps for analyzing a published dataset for CAF-I).

We have removed this phrase from the abstract.

d) Check spelling of all forms of the word (and processes related to) ubiquitin. There are multiple spellings/typos.

We have done this.

e) Features for the AI modeling are described as "chromatin features" but use features both within and outside of chromatin considerations. I would consider renaming.

We have renamed “chromatin features” as “chromatin and DNA features.”

f) There is a missed opportunity for more in-depth discussion of transcription factor contribution to potential pause sites and for discussion of potential RNA binding protein contributions.

We now added to the Discussion to raise these points (pg. 18).

[Editors’ note: what follows is the authors’ response to the second round of review.]

Essential revisions:1) The revised manuscript by Couvillion, Harlen, Lachance et al. is vastly improved. The authors have adequately addressed my main concerns. The only aspect that remains unclear to me concerns the fact that mutants for elongation factors such as dst1D and spt4D lead to decreased number of paused sites (Figure 4). As stated by the authors, this is unexpected since these factors are known to prevent (or help alleviate) pausing. Consistent with this expected behavior (and in apparent contradiction with the analysis shown in Figure 4), dst1D cells harbor a clear increase in Pol II density in the 5' region. This was highlighted in my initial review and the authors have addressed this by adding some speculations on page 12. This explanation, however, is vague and not compelling. One possible explanation would be that, in strains such as dst1D, pause sites are fewer but stronger. In this scenario, Pol II would pause less often but have a harder time getting out of the pause state in dst1D (and others) cells. Does the data allow testing this possibility? I feel that the manuscript would benefit from straightening that aspect.

The pause sites detected in NET-seq data are only the “stereotypical” ones, meaning that Pol II stops at those positions across many cells. Pol II is likely to pause at many other sites randomly, as has been observed for the *E. coli* RNA polymerase. It is unclear what percentage of Pol II pausing is expected to occur at the stereotypical sites and how that percentage changes upon the loss of regulatory factors. Nevertheless, as suggested by the reviewer, it is interesting to determine the overall pause strength at the stereotypical pauses, which we estimated by calculating the percentage of NET-seq reads at pause sites. The results of that analysis are shown in Figure 4 —figure supplement 1G.

This analysis shows that the Pol II density at stereotypical pause sites is lower in *dst1*∆ or *spt4∆* cells than in wild-type cells. It is important to keep in mind that there could be substantial pausing at noncanonical sites in these strains. Indeed, Pol II density near transcription pause sites increases in *dst1*∆ and *spt4∆* strains, reflective of more overall pausing in that region.

To increase clarity, we have added the following sentence on pgs. 10-11 to emphasize that our approach identifies only stereotypical pause sites and is blind to other Pol II pausing events. “Because NET-seq is performed in bulk on a population of cells, only the sites that consistently induce pausing are observed, and we refer to these as ‘stereotypical’ pause positions”. In addition, we added the following sentences to the end of the same paragraph.

“Stereotypical pause sites in NET-seq data represent loci where Pol II pauses in many cells and represent a fraction of the overall pausing by Pol II. The *E. coli* RNA polymerase pauses both at specific pause sites and randomly across a DNA template (Adelman et al., 2002; Neuman et al., 2003). Thus, Pol II is likely to similarly pause ubiquitously across gene bodies in noncanonical ways that would not lead to a detectable signal in NET-seq data. Nevertheless, the stereotypical pause sites identified here provide insight into the underlying features that induce Pol II pausing”.

2) For the most part my major concerns have been addressed. The reproducibility of the experiments was carefully assessed (Figure S5A & B) and the use of the irreproducibility discovery rate (IDR) with clear cutoffs sets clear quantitative standards for each dataset. It is clear that some of the knockout strains have a low overall impact on pausing and this is discussed through comparison of the median TSS pausing index.

We are pleased that our edits addressed the reviewer’s concerns.

3) Much of my major concern is with the use of a discussion of machine learning being used for prediction of pausing location. The machine learning section appears to more clearly provide new models for the contribution of different DNA/chromatin features to changes in pausing observed in any individual elongation factor deletion strain. This point can be addressed through writing to clarify what the machine learning analysis actually provides rather than what it could potentially do (predict pause sites) but appeared to fall short of.

All references to “predicting pauses” have been changed to “classifying pauses”. We also added the following sentence to pg. 15 to clarify the role of the machine learning analysis. “Which features contribute the most to the random forest classifier can help shape models for the molecular underpinnings of stereotypical Pol II pausing”.

4) I appreciate the care taken to address the concerns raised about the sense/antisense ratio analysis from the initial manuscript. This clarity of this section is much improved. I have a comment regarding this specific statement:"The factors whose deletions led to the largest increase in the antisense transcription were those involved in the regulation of histone acetylation, including members of the Rpd3S-Set2 pathway (Set2) and the major histone H4 acetyltransferase complex NuA4 (Eaf1), emphasizing the role of acetylation in antisense transcription (Carrozza et al., 2005; Churchman and Weissman, 2011; Krogan et al., 2003; Murray et al., 2015; Murray and Mellor, 2016)."For this statement, the deletion of an acetyltransferase will decrease acetylation whereas the deletion of members of the Rpd3S-Set2 pathway increase acetylation. As a consequence it is recommended to state "emphasizing the role of acetylation / deacetylation in antisense transcription" for clarity.

The suggested edit was made on pg. 8.